# Unified tumor growth mechanisms from multimodel inference and dataset integration

**Samantha P. Beik** [1], **Leonard A. Harris** [2,3,4], **Michael A. Kochen** [5], **Julien Sage** [6,7], **Vito Quaranta** [8,9], **Carlos F. Lopez** [9] *

1 Medical Scientist Training Program, Vanderbilt University School of Medicine, Nashville, Tennessee, United States of America, 2 Department of Biomedical Engineering, University of Arkansas, Fayetteville, Arkansas, United States of America, 3 Interdisciplinary Graduate Program in Cell & Molecular Biology, University of Arkansas, Fayetteville, Arkansas, United States of America, 4 Cancer Biology Program, Winthrop P. Rockefeller Cancer Institute, University of Arkansas for Medical Sciences, Little Rock, Arkansas, United States of America, 5 Department of Bioengineering, University of Washington, Seattle, Washington, United States of America, 6 Departments of Pediatrics, Stanford University, Stanford, California, United States of America, 7 Departments of Genetics, Stanford University, Stanford, California, United States of America, 8 Program in Chemical and Physical Biology, Vanderbilt University, Nashville, Tennessee, United States of America, 9 Department of Biochemistry, Vanderbilt University, Nashville, Tennessee, United States of America

* cflopezw@gmail.com

**Data Availability Statement:** All the code to reproduce the data analysis and the figures is open source and can be found in this GitHub repository: https://github.com/LoLab-MSM/Bayes-MMI. This

## Abstract

Mechanistic models of biological processes can explain observed phenomena and predict responses to a perturbation. A mathematical model is typically constructed using expert knowledge and informal reasoning to generate a mechanistic explanation for a given observation. Although this approach works well for simple systems with abundant data and well-established principles, quantitative biology is often faced with a dearth of both data and knowledge about a process, thus making it challenging to identify and validate all possible mechanistic hypothesis underlying a system behavior. To overcome these limitations, we introduce a Bayesian multimodel inference (Bayes-MMI) methodology, which quantifies how mechanistic hypotheses can explain a given experimental datasets, and concurrently, how each dataset informs a given model hypothesis, thus enabling hypothesis space exploration in the context of available data. We demonstrate this approach to probe standing questions about heterogeneity, lineage plasticity, and cell-cell interactions in tumor growth mechanisms of small cell lung cancer (SCLC). We integrate three datasets that each formulated different explanations for tumor growth mechanisms in SCLC, apply Bayes-MMI and find that the data supports model predictions for tumor evolution promoted by high lineage plasticity, rather than through expanding rare stem-like populations. In addition, the models predict that in the presence of cells associated with the SCLC-N or SCLC-A2 subtypes, the transition from the SCLC-A subtype to the SCLC-Y subtype through an intermediate is decelerated. Together, these predictions provide a testable hypothesis for observed juxtaposed results in SCLC growth and a mechanistic interpretation for tumor treatment resistance.

repository contains all the source code for each step in the analysis. Files needed to reproduce the figures without running the data analysis prior can be found at: https://doi.org/10.5281/zenodo.6671100 (TKO data files), https://doi.org/10.5281/zenodo.8002484 (RPM data files), and https://doi.org/10.5281/zenodo.8002506 (SCLC-A cell line data files). Download of these files and placement in the corresponding directories in the GitHub repository enables reproduction of Figs 1, 4–6, S1 and S3–S6.

**Funding:** S.P.B. was supported by NIGMS of the National Institutes of Health (NIH) [T32GM007347 and T32LM012412] (https://www.nih.gov/) and by the National Cancer Institute (NCI) [F30CA247078] (https://www.cancer.gov/); C.F.L. was supported by the National Science Foundation (NSF) [MCB 1411482] and NSF CAREER Award [MCB 1942255] (https://www.nsf.gov/); C.F.L. and V.Q. were supported by the National Institutes of Health (NIH) [U54-CA217450 and U01-CA215845] (https://www.nih.gov/). The funders had no role in study design, data collection and analysis, decision to publish, or preparation of the manuscript. The content is solely the responsibility of the authors and does not necessarily represent the official views of the National Institutes of Health.

**Competing interests:** The authors have declared that no competing interests exist.

## Author summary

To make a mathematical model, an investigator needs to know and incorporate biological relationships present in the system of interest. However, if the exact relationships between components of a biological system are not known, how can a model be constructed? Building a single model may include spurious relationships or exclude important ones. Therefore, model selection methods enable us to build multiple model hypotheses, incorporating various combinations of biological features and the relationships between them. Each biological feature represents a distinct hypothesis, which can be investigated via model fitting to experimental data. In this work, we aim to improve upon the information theoretic framework of model selection by incorporating Bayesian elements. We apply our approach to small cell lung cancer (SCLC), using multiple datasets, to address hypotheses about cell-cell interactions, phenotypic transitions, and tumor makeup across experimental model systems. Incorporating Bayesian inference, we can add into model selection an assessment of whether these hypotheses are likely or unlikely, or even whether the data enables assessment of a hypothesis at all. Our analysis finds that SCLC is likely highly plastic, with cells able to transition phenotypic identities easily. These predictions could help explain why SCLC is such a difficult disease to treat and provide the basis for further experiments.

## Introduction

A mechanistic understanding of biological processes that explains causal input-output relationships and predicts population behaviors [1] remains a central challenge to all areas of quantitative biology. Mathematical models have become an established practice to specify precise relationships within a biological system, [2] and thereby hypothesize, and subsequently test, the existence of these relationships. For example, multiple mechanistic models of apoptosis execution have been formulated to explore the nature of biochemical interactions that lead to cellular commitment to death, demonstrating that careful model design and suitable data can lead to important biological insights [3–5]. A more challenging situation emerges when models are formulated for biological processes that are poorly defined or understood, leading to multiple competing, and often juxtaposed mechanistic explanations for a given biological process. For example, in Small Cell Lung Cancer (SCLC), a study of circulating tumor cell-derived xenografts showed that relapsed inflammatory, non-neuroendocrine subtypes could act as a stemlike population ("source") [6], but archetype analysis of a genetically engineered mouse model tumor showed the non-neuroendocrine subtype SCLC-Y acts as an end-state ("sink", rather than source) [7]. Therefore, continued exploration of the hypotheses generated from these works can help elucidate the differences between these and other potential explanations for tumor growth mechanisms.

This phenomenon where multiple mechanistic hypotheses are concurrently proposed but must be assessed with limited data is not restricted to quantitative biology but common to other fields with similar data availability limitations such as ecology [8] climatology [9], and evolutionary biology [10], to name a few. To address this challenge, methods such as model selection and multimodel inference have been proposed using information theoretic scoring techniques such as Akaike Information Criterion, (AIC) with success in differentiating between models of different mechanistic hypotheses, but limited benefits for model averaging given that AIC scores do not inherently describe whether a model or features within are informed by the data. More recently, the use of AI and machine learning approaches has given

impetus to causal relationship inference [11] but these relationships remain difficult to elucidate, thus underscoring the need for both novel tools for hypothesis exploration, and tools that can be used with rigor in the face of missing data.

The model selection process involves candidate model evaluation from a superset of plausible models, relative to a given experimental dataset. The most common approach for model selection is to consider a "best" model as comprising the most relevant variables that capture important mechanistic aspects of the explored process, while excluded variables capture process features that are less relevant for the question being explored. However, variables throughout all candidate models can contribute to knowledge about the overall system [12]. In the cases where data is simply less informative for a given set of hypotheses, uncertainty will remain about what constitutes a "best" model [13], necessitating approaches such as model averaging, where parameter values can be weighted by model probability and then combined into a distribution of likely values [12,14]. Unfortunately, for information theoretic applications of model averaging, this probability must be weighted and summed across all possible models, which are often not possible to enumerate exhaustively for all parameter combinations.

To address the challenge of employing multimodel inference approaches in the context of biological processes where models cannot be exhaustively enumerated, data may not inform all model evaluation, and model averaging across all models is desired to learn about the system of interest, we present a Bayesian multimodel inference workflow (Bayes-MMI), which combines Bayesian inference with model selection and model averaging. Bayesian model selection, i.e. calculating the likelihood of a model in comparison to data, and Bayesian model averaging, determining an "averaged" model or averaged parameter values for predictions, have previously been applied across multiple disciplines [13,14]. To our knowledge, our approach is novel in its incorporation of these methodologies specifically to explore biological mechanistic hypotheses represented by inclusion or exclusion of species and behaviors. Application of Bayesian principles in turn reveals the extent to which data informs a given model and its constitutive species, parameters, and overall behaviors.

For small cell lung cancer (SCLC), our system of interest, we integrate the most suitable available datasets and published theories of SCLC cellular biology to identify a set of mechanistic hypotheses for SCLC tumor growth. We test the resulting thousands of candidate population dynamics models via nested sampling, comparing candidate model output to tumor steady-state data, applying the principles of model selection and model averaging for a principled and comprehensive assessment of SCLC mechanistic hypotheses. While fitting kinetic parameter rates in a model to predict rates of biological actions that correspond to model parameters is common, here we only evaluate general trends across varying categories of mechanistic models and instead focus on evaluating the probability of mechanistic hypotheses given the data at hand. Estimating these probabilities, we generate an interpretation of SCLC tumor growth: highly likely non-hierarchical phenotypic transitions indicating SCLC subtype plasticity, and less likely cell-cell interactions that affect the rate of phenotypic transitions across subtypes. We show how certain aspects of the SCLC model, such as phenotypic transitions and cell-cell interactions related to these, are well informed by the available data, but other aspects, such as tumor initiation and growth rate effects, are not informed. Our approach is generalizable to other biological systems, and as such we advocate for a shift away from considering the "best" model and variables within, and propose a move toward Bayesian-driven multimodel inference for a probabilistic understanding mechanistic biological processes.

## Results

### Bayesian inference efficiently infers parameter inclusion in a "ground truth" example model

To demonstrate the advantage of a Bayesian approach to multimodel inference, we employ a simple model selection and model averaging example as shown in Fig 1. With generated "ground truth" data based on the work of Galipaud and colleagues [15], we employ Akaike Information Criterion (AIC) and Bayesian multimodel inference (Bayes-MMI) to compare the AIC vs. Bayes-MMI evaluation of parameter inclusion in a model representing the "ground truth" data.

Model selection aims to capture a balance between optimizing a model to match a given dataset exactly, while also having the fewest terms required to do so [12,16]. In biological investigations, model selection typically employs information criteria metrics such as the Akaike Information Criterion (AIC) [12,16], or less often, Bayesian methods such as calculating the marginal likelihood [17]. AIC scores are based on the single highest likelihood value found during parameter optimization, while the marginal likelihood incorporates all likelihood values from parameter estimation and represents essentially their average over the prior space (we refer interested readers to S1 Text **Note A** for further detail including a didactic example detailing the theoretical background behind AIC, marginal likelihood, and associated probabilities). Model averaging enables evaluation of individual hypotheses via model pieces (as described in Fig 1A) by using the outcomes of model selection. Sums of AIC weights (SW) (for theory and calculation see S1 Text **Notes A.3, A.5, A.6**) are *treated* as probabilities that a model variable is part of the "true" model, while Bayesian model averaging (BMA) *calculates* probabilities using Bayes' Theorem (see S1 Text **Notes A.2, A.5, A.6**). In biological applications, a variable with high SW would indicate that a biological feature plays an important role in the system. Having a high posterior probability would also indicate that a biological feature plays an important role, but posterior probabilities allow for additional details such as whether the data used for parameter optimization informed that model variable (see S1 Text **Note A.6**).

While AIC has been successful in differentiating between models in a set of candidates [12,14,16], problems have been noted in moving from ranking models to evaluating model variables via AIC-based SW [18,15], such as in the example we adapt here. Using code provided in [15], we generated "ground truth" data from four variables $x_{1-4}$, all with differing correlations with the response variable $y$, including perfect correlation between $x_1$ and $y$ and no correlation between $x_4$ and $y$ (Pearson correlation coefficients 1.0 and 0.0, respectively) (S1 Text **Note A.7**). Shown in Fig 1B is the set of 16 linear regression models that may represent this "ground truth" data; based on the data, $x_1$ should be included in the optimal model, and $x_4$ should be excluded. To test how Bayes-MMI compares to AIC-based SW methods in evaluating inclusion/exclusion of model variables, we perform a nested sampling analysis via PyMultinest, [19–21] (see Methods). Using the marginal likelihood, we calculate candidate posterior probabilities to perform our own ranking of models (Fig 1C; S1 Text, **Table B**). We calculate AIC corrected for small sample sizes (AICc) SW and variable posterior probabilities (Table 1, Fig 1D; S1 Text, **Table C**) using equivalent prior probabilities of variable inclusion. Overall, we find that both our own marginal likelihood values as well as the AICc results follow similar trends as those found by Galipaud and colleagues (S1 Text, **Tables A and C; Note A.7**).

AICc-based SW overestimates the probability of $x_4$ inclusion in the "true" model, as demonstrated in Fig 1C and 1D and Table 1. SW for $x_4$ in [15] is 0.37, larger than would be expected given that $x_4$ has 0 correlation with $y$. SW for $x_4$ in *our* AICc analysis is 0.25, while the posterior probability that the inclusion of $x_4$ is supported by the data is 0.09 (Fig 1D; S1 Text, **Table C**). There does not appear to be an accepted threshold over which the SW for a variable

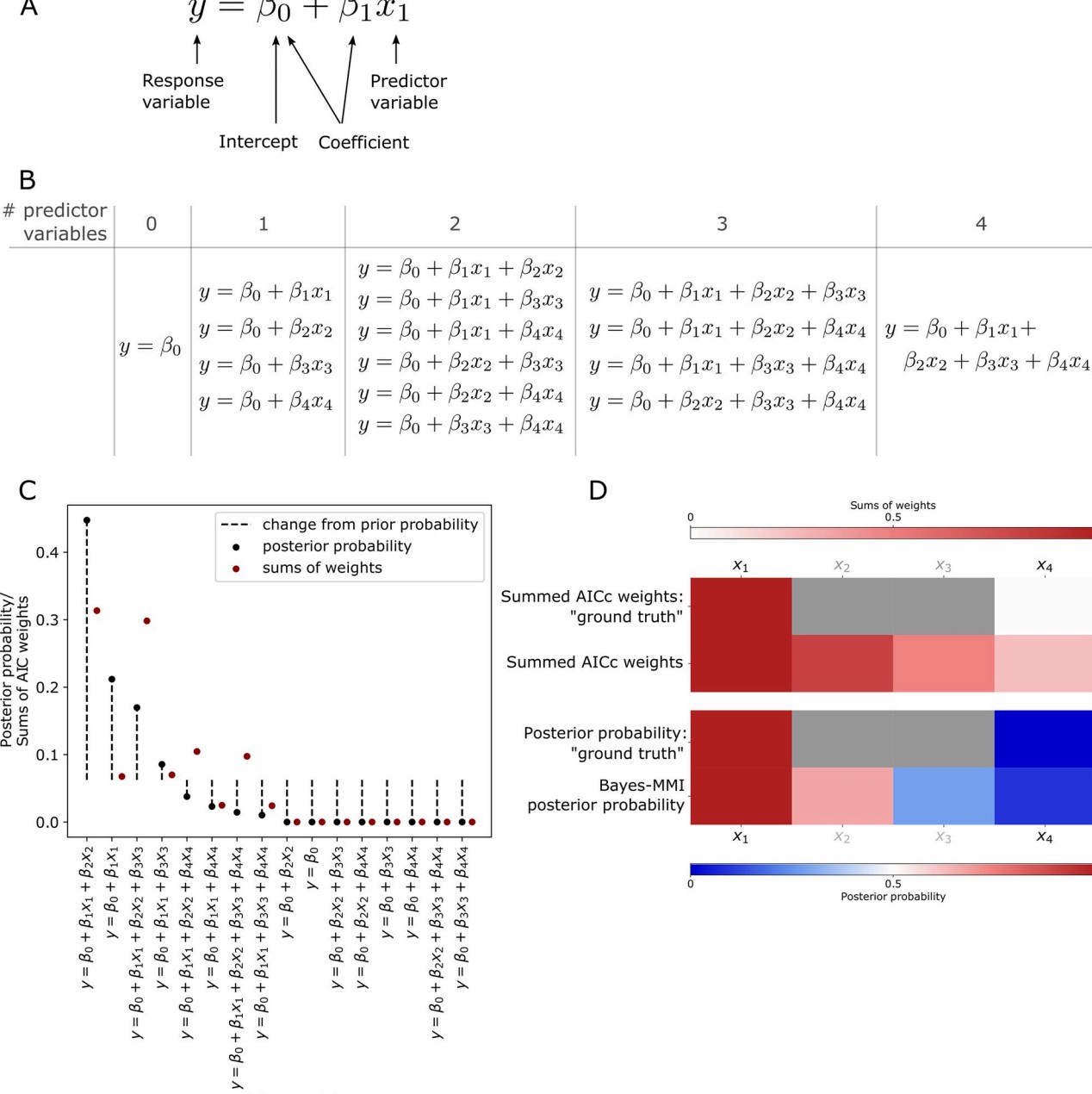

**Fig 1. Bayesian inference better assesses parameter inclusion in the "true" model.** (A) Aspects of linear regression model assessed by model selection and model averaging. (B) Candidate linear regression model set for multimodel inference analysis and comparison to AICc in our analysis of the example problem from Galipaud *et al.*, 2014 [15]. (C) Results from our analysis of candidate models in (B) using data generation code from Galipaud *et al.*, 2014 [15]. Candidate models are arranged along the x-axis by posterior probability. Each posterior probability from our analysis is compared to the AICc weight from our analysis. (D) Heatmap for each model averaging analysis based on data generation and candidate models in [15]. Top two rows are summed AICc weights (SW), where SW is a value starting at 0 and weights are summed until the final SW value; thus, the top color bar corresponding to these two rows moves from 0 (white) to 1 (dark red). First row, the idealized summed AICc weights based on generated "ground truth" data (with Pearson correlations between $x_2$ and $y$ and $x_3$ and $y$ different than 0.0 or 1.0, we cannot be sure what the "true" SW should be); second row, summed AICc weights from our analysis. Note the large difference between ideal SW for $x_4$ (zero) and its calculated SW (0.25; see Table 1). Bottom two rows are posterior probabilities from nested sampling and Bayes-MMI, where before nested sampling, prior probabilities are at 0.5 (white in the bottom color bar) and color in the heatmap represents the probability of the variable's inclusion in the "true" model (darkest blue for 0% probability, darkest red for 100% probability in the bottom color bar). Deeper colors indicate a larger deviation from the prior. Third row, the idealized probability based on "ground truth" data (similarly to SW, Pearson correlations between 0 and 1 for $x_2$ and $x_3$ mean we cannot be sure what "true" posterior probability should be); fourth row, posterior probability from our Bayes-MMI analysis. Note the closer correspondence between ideal posterior probability for $x_4$ (zero) and its marginal-likelihood-derived probability (0.09; see Table 1) than between ideal SW and calculated SW.

**Table 1. SW and posterior probability calculations for each model variable.**

| Variable | SW: Galipaud *et al.* 2014 [15] | SW: this manuscript | Prior probability | Bayes-MMI posterior probability |
|---|---|---|---|---|
| $x_1$ | 1 | 1 | 0.5 | 1 |
| $x_2$ | 0.94 | 0.81 | 0.5 | 0.67 |
| $x_3$ | 0.37 | 0.49 | 0.5 | 0.28 |
| $x_4$ | 0.37 | 0.25 | 0.5 | 0.09 |

Prior probability for a variable is set at 0.5, meaning a variable's prior probability for can be calculated per candidate model by dividing 0.5 by the number of models in which the variable appears. Prior probability values only impact posterior probability scores and not SW calculations.

indicates it should be included in a model, thus practitioners choose a threshold on a per-case basis [15]. However, using Bayesian principles, we see the posterior probability of $x_4$ inclusion at 0.09 has decreased to a large extent from its prior probability of 0.5 (Table 1). We view 0.09 as a more reasonably low probability than 0.37 or 0.25, given that $x_4$ is not at all correlated to *y*. (Please see S1 Text, **Notes A.7, B and C** for additional analyses performed on this example.)

The results from this simple example suggest that Bayesian analysis-based methods on the marginal likelihood exhibit improved performance for model selection, specifically as it pertains to inclusion or exclusion of variables or model terms. We apply these principles to SCLC tumor growth, our system of interest, and employ multimodel inference (MMI; model selection and model averaging together) in the context of Bayesian statistics. There have been biological investigations using MMI approaches [22,23] but, to our knowledge, this work is the first application of Bayes-MMI to cell population dynamics models. Using MMI to assess the posterior probability of different model variables is comparable to Bayesian variable selection, which in biomedicine has been used to determine genetic loci associated with health and disease outcomes in linear models [23]. We find using a Bayesian approach to MMI results in probabilities that biologically relevant model features are (or are not) supported by the data. Such an approach is likely relevant to any cancer or developmental biology application and can be used to investigate model variables even in the context of limited or uncertain data.

## Existing datasets yield multiple hypotheses in SCLC tumor growth mechanisms

Small cell lung cancer (SCLC) has been denominated a recalcitrant tumor, signifying that relapse after treatment is commonplace and survival prognosis is typically poor. SCLC comprises ~15% of all lung cancer cases worldwide and results in ~200,000 deaths annually with a 5-year survival rate of less than 10% [24]. Intratumoral heterogeneity is hypothesized to be the main contributor to the natural history of this disease and its morbidity and mortality [24–26]. SCLC tumors comprise a mix of functionally distinct subtypes of interacting cells, [27–29], most notably neuroendocrine (NE) and Non-NE. As shown in Fig 2A, SCLC populations comprise a collection of cellular subtypes within a tumor, identified by differential expression of transcriptional regulators [25].

The overall goal in this work is to computationally explore tumor growth mechanism hypotheses in SCLC. Tumor features that emerge as highly supported by data about the growth mechanism could be used to predict differences in growth across tumors of different genetic backgrounds, responses to *in silico* treatment, or even predict patient-specific tumor behavior after various treatments. Unfortunately, these goals are currently hypothetical, because to build one SCLC model that could be used for these purposes, one would need a unified understanding of the SCLC tumor as a system, and knowledge of SCLC currently exists as

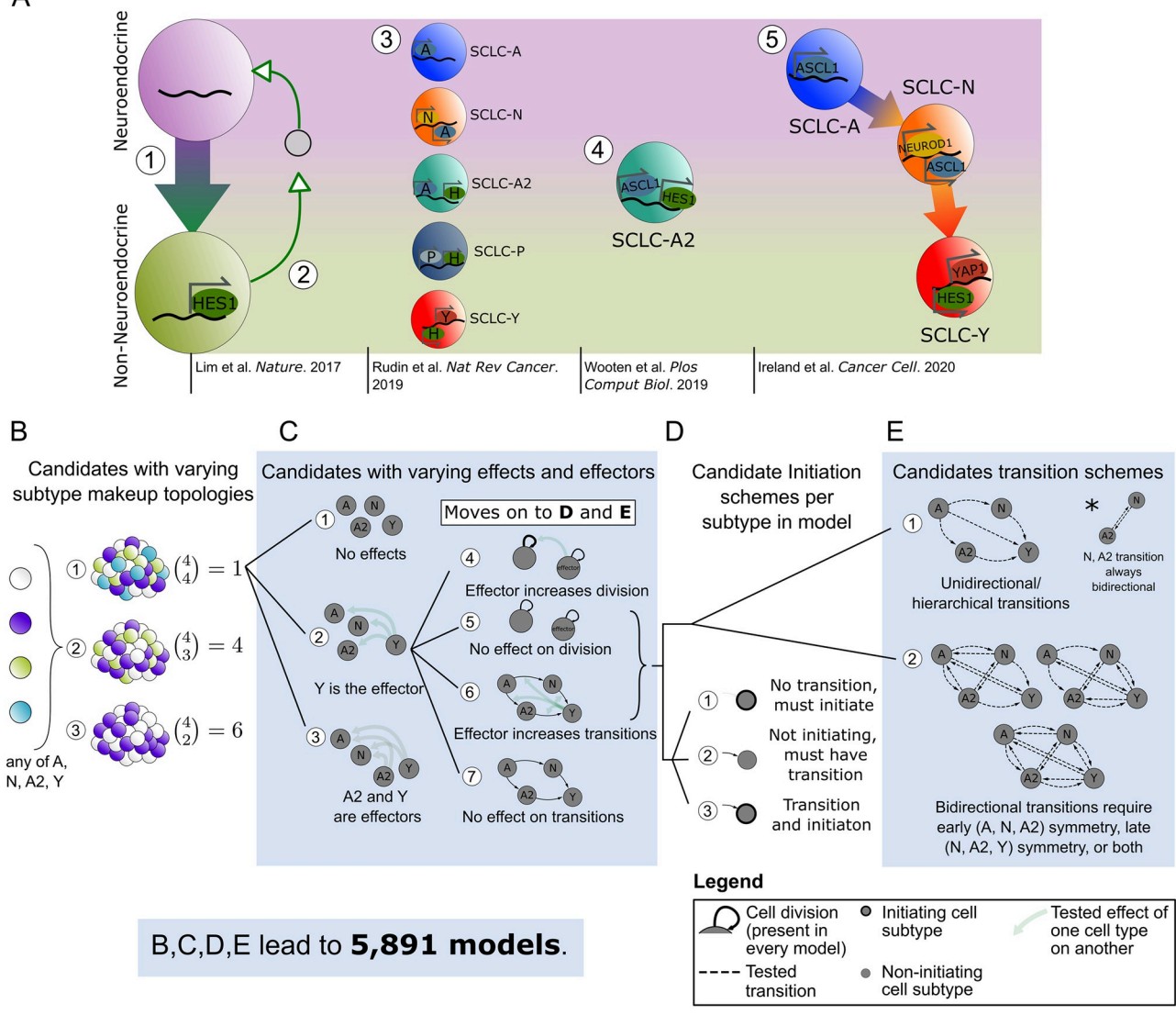

**Fig 2. Conclusions and hypotheses from literature build mechanistic hypothesis exploration space for tumor growth and development.** (A) Synthesis of what is currently known about SCLC subtypes, which have been divided into two overall phenotypes, neuroendocrine (NE) and Non-NE, and then further classified into subtypes based on transcription factor expression. [1] NE SCLC cells, which do not express HES1, transition into Non-NE cells, which do. [2] HES1+ cells release unidentified factors (gray circle) that support viability and growth of HES1- cells, and the two HES1+ and HES1- populations grow better together rather than separately. [3] Consensus across the field led to labeling SCLC phenotypic subtypes by the dominant transcription factor expressed in that subtype. [4] Subtype with transcriptional signature intermediate between NE and Non-NE, named SCLC-A2. [5] Phenotypic transitions occur in a hierarchical manner from SCLC-A to SCLC-N to SCLC-Y cells. (B)-(E) Candidate model examples representing SCLC biological hypotheses (Table 1). Here we indicate schematically how a population dynamics model can represent each biological hypothesis, as well as denote how the set of candidate models is built combinatorially, in order of (B)-(E). (B) Model topologies constructed with 2 + subtypes, with number of combinations per number of subtypes. There are 11 options total, and each of these move forward to choose one effect option from (C [1], [2], or [3]). (C) Subtype effect schema, where there are different effectors between candidates and different affected cellular actions. If there are effects (C [2] or [3]), model behaviors affected are chosen (choose (C [4] & [6], [4] & [7], or [5] & [6]). Whether effects present (C [2], [3]) or not (C [1]), the candidate moves forward to choose initiating subtype(s): each subtype in the model must follow (D [1], [2], or [3]) and corresponding transition schemes (E [1], [2]). (D, E) Initiation schemes (D) and potential transition schemes (E), where all subtypes in topology must be accessible either as initiating subtypes or via transitions (D), unidirectional transitions are those that follow a hierarchy (E, top left), and bidirectional transitions must be symmetrical when present (E, top right and bottom). A: ASCL1, Achaete-scute homolog 1; N: NEUROD1, neurogenic differentiation factor 1; H: HES1, Hes Family BHLH Transcription Factor 1; P: POU2F3, POU class 2 homeobox 3; Y: YAP1, yes-associated protein.

nonoverlapping conclusions and hypotheses. We summarize the current knowledge about SCLC tumor growth mechanisms, highlight potential knowledge gaps, and refer interested readers to [25] for a comprehensive review of recent SCLC literature beyond that noted here.

Multiple SCLC subtypes have been identified depending on the experimental model studied, shown in Fig 2A, as SCLC-A, N, A2, and Y [25]. Other experiments have led to additional proposed phenotypic subtypes, including canonical subtype SCLC-P, but these were not included in our analysis because they were not shown to be present in our datasets used [6,29–32]. Our previous work aimed to identify whether all subtypes may be present in a tumor or if only a subset are present, with the result that tumors can be composed of one, multiple, or all subtypes tested [33]. A comprehensive account of initiating SCLC subtype(s) (cell(s) of tumor origin) has not been made, but multiple have been hypothesized in [33–35].

Studies *in vitro* and *in vivo* have suggested that Non-NE subtype(s) support growth of NE subtypes [28] (Fig 2A [2]), including vasculogenic mimicking SCLC cells having such supportive effects [29]. The presence of NE subtypes has a dampening effect on Non-NE growth [36]. Recent work has shown that the HES1-positive (Non-NE) cells supporting NE subtype growth [28] may have upregulated YAP1 [37] (Fig 2A [2]), and are likely SCLC-Y; otherwise, the referenced studies were completed before the adoption of the canonical subtypes SCLC-A, N, A2, P, and Y, and so it is unclear which of these exactly contribute such effects in each case.

NE cells may undergo a transition a toward HES1+ (likely YAP1+) identity, (Fig 2A [1]) which modulates these Non-NE cells' sensitivity to anticancer drug treatment [28]. Other work found that SCLC-A subtype cells can transition to the SCLC-N subtype and from SCLC-N to SCLC-Y [34] (Fig 2A [5]). Without the ability to undergo a transition toward a more Non-NE phenotype, tumors were smaller and less aggressive; however, this study did not assess Non-NE or SCLC-Y sensitivity to anticancer drugs [34]. These two landmark studies assessing phenotypic transitions do not assess the same phenotypic transition pathway and thus we cannot compare intermediates, although we hypothesize that the transitions begin with SCLC-A and it seems reasonable to assume the hierarchical pathway ends in SCLC-Y. While our investigations support that SCLC-Y acts as an end state for phenotypic transitions, [7] another study identified that Non-NE subtypes may have stemlike potential [6], which contrasts with Non-NE or SCLC-Y acting as the end of the hierarchical pathway.

## Multiple mechanistic hypotheses emerge from existing data

Considering the aspects of SCLC tumor growth observed in the previous section, it is clear that no one model exists that could easily recapitulate all datasets. To address this challenge, we explored mechanistic hypotheses in the realm of tumor initiation and composition, phenotypic transitions and their hierarchy, and subtype-to-subtype effects (Table 2). Each point in Table 2 represents a biological hypothesis or theory of SCLC. To select which of these multiple hypotheses to include in a mechanistic model of SCLC without additional findings would introduce bias into the modeling process.

Instead, we can address these questions computationally, by including or excluding these behaviors across multiple mechanistic models and evaluating whether model behaviors recapitulate SCLC data; that is, we turn to model selection [12,14]. Using model selection, we generate a set of candidate models, each of which may include one biological hypothesis (one bullet point in Table 2), multiple hypotheses, or all hypotheses represented by in Table 2. In this way, the set of candidate models will represent SCLC hypotheses individually or in combination, and thus each hypothesis will be evaluable mathematically in the model selection workflow.

**Table 2. Existing data pertaining to SCLC intratumoral heterogeneity and communication.**

Tumor composition
- Neuroendocrine-classified (NE) subtypes are A (ASCL1+) and N (NEUROD1+), with non-neuroendocrine-classified (Non-NE) subtype Y (YAP1+) and P (POU2F3+). [25]
- Previous work predicted subtype A2, and SCLC-A, SCLC-A2 and SCLC-N have been seen to express ASCL1 [25,28,33,34]. SCLC-A2 expresses HES1. [33]
- HES1-positive TKO tumor cells (Non-NE) have YAP1 upregulated. [37]
- It is unclear whether A2 is more NE or Non-NE in character.
- Tumors can be made up of one or more of these subtypes. [25,33]
- Some subtype combinations have been experimentally verified and others have been predicted using CIBERSORT. [25,33,34]
- We do not see SCLC-P in our previous subtype deconvolution. [33]

Evidence of phenotypic transitions
- TKO tumor cells sorted for HES1-negativity (NE identity) become HES1 positive (Non-NE identity) when plated with Notch ligand DLL4. [28]
- *Ex vivo* culturing of *in situ* RPM tumors results in histologic and transcriptional phenotypic changes from NE to Non-NE gene expression over time. [34]
- Transitions between A and A2, A2 and N have not been studied.
- No evidence of SCLC-Y transition to NE identity. [28]

Subtype-to-subtype effects
- Cell viability and division are increased when HES1-negative cells are plated with HES1-positive cells, compared to HES1-negative cells only. [6,28,34,36,38]
- NE cells suppress Non-NE cell division. [39]
- Application of conditioned media from TKO-derived HES1+ (Non-NE) cell culture or of isolated exosomes TKO-derived from HES1+ cells results in a morphological change in TKO-derived NE cell line KP3.*

---

* *Personal communication (Alissa Weaver, Vanderbilt University)*. ASCL1, Achaete-scute homolog 1; NEUROD1, neurogenic differentiation factor 1; POU2F3, POU class 2 homeobox 3; YAP1, yes-associated protein; HES1, Hes Family BHLH Transcription Factor 1. TKO, $p53^{fl/fl}$;$Rb^{fl/fl}$;$p130^{fl/fl}$ tumors [28]; RPM, $Rb1^{fl/fl}$;$Trp53^{fl/fl}$;Lox-Stop-Lox [LSL]-$Myc$T58A tumors [36]

For our tumor growth mechanism exploration, we interpret tumor topology, initiation, potential subtype behaviors, etc., as features in candidate models (Table 2). We define *model variables* as representations of species in the model (e.g., "subtype A"), and *model terms* as qualitative actions in the model (e.g., "subtype A cell division"), whose rates are denoted by *kinetic parameters* (e.g., "subtype A division rate") (S2 Text **Note B**).

Studies of SCLC as well as other tumors indicate that multiple phenotypes may emerge in a tumor due to perturbations, even though fewer phenotypes are seen in their absence [34,40–46]. In SCLC, at least three, and as many as five, cell subtypes have been observed when perturbing the tumor via treatment, time, or environment change [6,28,33,34,36,38]. To perform our analysis with minimal bias, fully account for all possible tumor subtype compositions, and explore an exhaustive set of possible biological explanations, we explored models comprising between two and four subtypes per model (Fig 2B). In addition, we included all possible cell subtype interactions. Growth supportive effects and transition-inducing effects (Fig 2C) (and growth dampening effects, not shown) are included in some candidate models where, e.g., presence of an effector (supportive cell subtype) increases the rate of growth of a subtype it affects (supports). Subtype A2 has expression features of both NE and Non-NE cells [33], including expression of ASCL1 (seen in NE cells) and HES1 (seen in Non-NE cells) and we therefore assigned A2 NE features in some candidate models and Non-NE in others (Fig 2C).

To compare a hierarchical system, where a cancer stem cell (CSC) can (re)populate a tumor, and non-hierarchical systems in which phenotypic transitions can occur among multiple or all SCLC subtypes, we include candidate models with several different potential phenotypic transition schemes. Thus, the set of candidate models considered include models without phenotypic transitions, models with transitions that reflect hierarchical transitions observed experimentally [28,34], and models with reversible transitions, i.e., high plasticity (Fig 2D and 2E). Unidirectional transitions stemming from one cell subtype indicate a potential CSC, while bidirectional transitions from multiple subtypes indicate phenotypic plasticity. We additionally include tumor initiation from one cell of origin *vs.* multiple. Tumor initial conditions are not fitted, but set at 100 cells total, divided by whichever cells have been designated as initiating in the candidate model (S2G Fig). Thus, a certain set of model variables and terms result in not just one candidate model, but $2^n$-1 (where *n* is the number of subtypes in the model topology) candidates to account for all possible initial conditions (Fig 2D and S2G Fig).

To ensure we built a comprehensive set of candidate models that enable exhaustive exploration of biologically relevant hypothesis space, we combined the potential SCLC behaviors (Fig 2 and Table 2) with prior knowledge about mechanistic behavior of tumor populations [47–52]. For example, if there is indeed plasticity in the system, it is likely to be shared among subtypes, leading to symmetrical bidirectional phenotypic transitions across the model (Fig 2E). We therefore expect that all plausible SCLC tumor growth mechanisms are represented in our candidate model hypothesis space to the best of our knowledge.

While we consider this an exhaustive exploration of the *biologically relevant* hypothesis space, it does not exhaustively explore the entire potential model hypothesis space. We have 44 tunable parameters to choose from in building a candidate model (see S1 Table for rationale behind parameter prior decisions, and S3 Fig for graphical representation of prior parameters), and 15 initial conditions (S2G Fig). Were we to exhaustively search model space, this would result in approximately $15*2^{44}$, around 260 trillion, models to test. Accounting for all the biological possibilities noted above led to a set of 5,891 unique candidate models, 0.000000002%, or approximately two billionths of a percent, of the entire model hypothesis space; each of the resulting 5,891 models represents a possible SCLC tumor growth mechanistic hypothesis.

## Bayesian exploration of candidate population dynamics models using experimental data

We use multiple datasets to identify consensus behavior of SCLC and provide a unifying model of tumor growth mechanisms broadly supported by available data (Fig 3A). These datasets include two genetically-engineered mouse models (GEMMs), the triple-knockout (TKO) model ($p53^{fl/fl}$;$Rb^{fl/fl}$;$p130^{fl/fl}$ tumors [28]; equivalent to the RPR2 GEMM [34,36,53]), and the RPM model ($Rb1^{fl/fl}$;$Trp53^{fl/fl}$;Lox-Stop-Lox[LSL]-$Myc^{T58A}$ [36]), and cell lines from the Cancer Cell Line Encyclopedia (CCLE) [54] made up largely of the SCLC-A subtype determined in [33] (Fig 3A and S1 File). While the SCLC-A cell lines are samples from varying (human patient) genetic backgrounds, we estimate a similar genetic background based on the similar tumor composition. Previous work from our labs suggests that the tumor genetic background dictates the potential behaviors of phenotypic subtypes within a tumor population, [55–57] and so these three datasets represent three potential tumor cell suites of behaviors. Using this data provided us with proportions of tumor samples assigned to SCLC subtypes, using the same gene signatures across all samples, automatically determined by CIBERSORT from samples of CCLE SCLC cell lines [54] and consensus clustering class labels [33]. We consider this preferable to our own *ad hoc* decisions of individual cell subtype identity necessary to assign the required subtype proportions of tumors had we used newer, available single-cell RNA

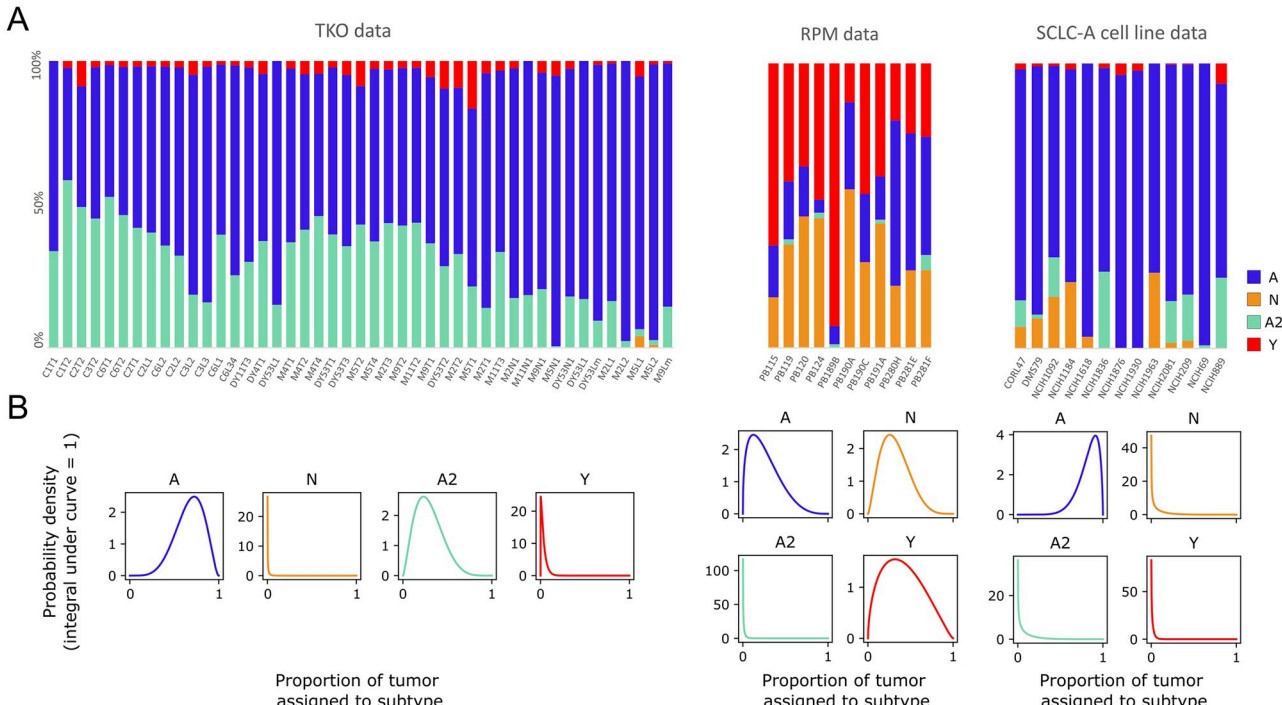

**Fig 3. Population composition data and probabilistic representation.** (A) CIBERSORT deconvolution of TKO and RPM genetically engineered mouse model (GEMM) samples (previously published) as well as SCLC-A cell line samples. CIBERSORT was performed on bulk RNA-sequencing data. (B) Probabilistic representation of tumor proportion based on mean and standard deviation of proportions across samples within an experimental model; these distributions were then used for fitting models to data. TKO, $p53^{fl/fl};Rb^{fl/fl};p130^{fl/fl}$ tumors [28]; RPM, $Rb1^{fl/fl};Trp53^{fl/fl};$Lox-Stop-Lox[LSL]-$Myc^{T58A}$ tumors [36]; SCLC-A cell lines, a subset of SCLC cell lines from the CCLE [54] that we previously assigned as representative of tumors made up largely of the SCLC-A subtype [33].

sequencing data. The different datasets represented in Fig 3A demonstrate differing SCLC tumor makeup, according to the experimental model employed in the study.

To explore the roles of phenotypic heterogeneity and cellular behaviors (cell-cell interactions, phenotypic transitions) on SCLC tumor growth dynamics, we used population dynamics modeling, building on our previous work [55]. Population dynamics models employ a mathematical description of the dynamics within and between heterogeneous subpopulations in an overall population [58,59]. With such models, researchers can mathematically simulate population growth over time and investigate growth dynamics inherent in the simulations (S2 Text **Note A**).

With 5,891 candidate models, (Fig 2B–2E) we aimed to determine which one(s) could best represent the SCLC system [12,16]. Information theoretic approaches for model selection using the Akaike Information Criterion (AIC) have been used in prior work with success in determining the optimal model or subset of models, but do not yield a much-needed Bayesian statistical understanding of the data. For a didactic demonstration of this, we refer the reader to S1 Text **Note A**. We therefore employed the marginal likelihood as a more principled means for model ranking and model averaging.

Prior work has estimated the marginal likelihood for kinetic model fitting using thermodynamic integration [17]. In this work we instead use nested sampling, [60], which is computationally more efficient and has fewer limitations with regard to the shape of the probability space traversed during evidence calculation [19–21]. The nested sampling method was run once for each of the 5,891 candidate models on each of the three experimental datasets,

amounting to 17,484 potential interpretations of tumor growth mechanisms. The average fitting time for each model was ~19 wall-clock hours, thus necessitating high-performance computing for a complete parameter space exploration of the candidate models.

Each model is thus optimized to our datasets via nested sampling, which explores the full volume of the likely parameter space. Each point in parameter space represents a set of possible parameter values (S3 Fig). At each of these points, nested sampling assigns a likelihood value for how well that set of parameter values fits the data. On completion of the algorithm, the output includes the highest-likelihood parameter values. Since each tested point in parameter space is a set of parameter values, the highest-likelihood values for a model are returned by the algorithm as a list of parameter sets (S4 Fig). Returning a list of parameter sets rather than one top-scoring set already incorporates Bayesian methodology into the process—each individual parameter has multiple best-fitting values, which can be interpreted as a distribution of parameter values [17]—but with nested sampling we add yet more Bayesian methodology. Having assigned a likelihood to every point in parameter space, nested sampling uses these to calculate one overall likelihood per model, the marginal likelihood, which takes into account parameter fit as well as model simplicity (number of parameters). For more detail on how the marginal likelihood is calculated to incorporate both model fit and size, see S1 Text **Note A** and Methods. Finally, with marginal likelihood values for each model in the candidate set, and the candidate model set representing the full hypothesis space with all potential SCLC population dynamics models, we can calculate a probability. Summing the marginal likelihood values, and dividing each individual marginal likelihood by these, results in a model posterior probability, representing a change in probability from pre-model fitting (all models with equal prior probability) to post-model fitting (see S1 Text **Note A**). We are then able to compare model probabilities and additionally perform model averaging to evaluate kinetic parameter value distributions and probabilities of model variables and terms.

## A small subset of candidate tumor growth models is supported by experimental data

We aim to perform multimodel inference, comparing model probabilities and parameters using model selection and model averaging, to hypothesize which relationships within the SCLC system are most informed by the data. As noted previously, each model represents the combination of multiple biological hypotheses, and we begin by determining how well our candidate SCLC models best represent SCLC.

In our linear regression example, we investigate which predictor variables, with their fitted coefficients, best match the response variable (Figs 1A and 4A). To evaluate the behavior of SCLC, we use kinetic models, with cell behavior represented by ordinary differential equations (ODEs) (see Methods; S2 Text **Notes A and B**), which for multimodel inference involves investigating which variables and which model terms, with their fitted kinetic parameters, best match the data (Fig 4A–4C).

In a Bayesian model selection approach, a more likely model comprises a higher proportion of the probability of the candidate model space (Fig 4D–4F). After nested sampling, our results indicate the highest-scoring model for each dataset is ~$10^{19}$ times more likely than the lowest-scoring model, and ~$10^3$ times more likely than the median scoring model. For reference, the smallest comparison between models that is considered significant is $10^{1/2}$ [61] (S1 Text **Note A.6**).

Performing nested sampling on all candidate models did not yield a unique best-fitting model for any dataset (Fig 4D–4F). We therefore leveraged a multi-model inference approach and calculated a confidence interval (CI) representing a set of best-fitting models per dataset.

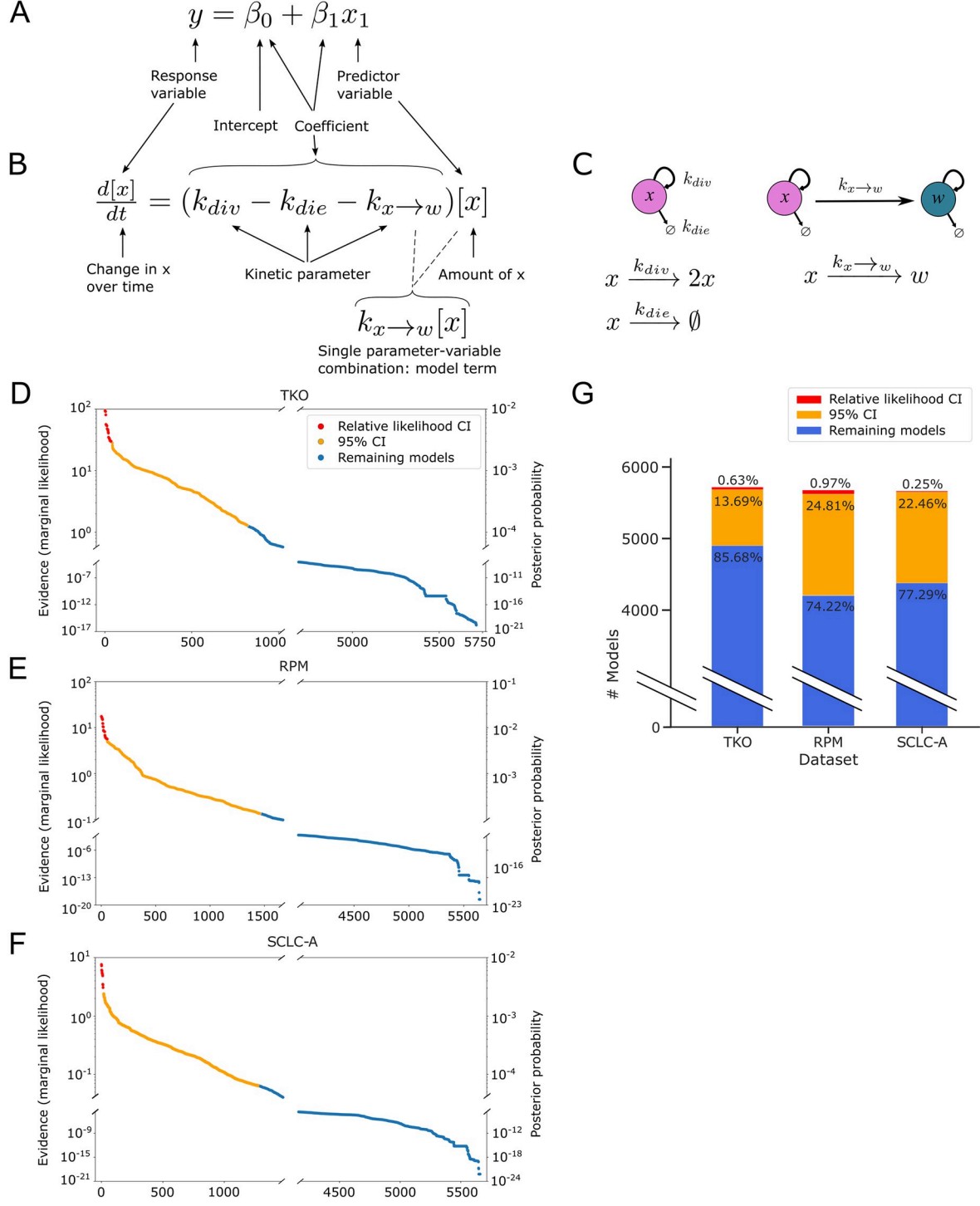

**Fig 4. Fitting to data and assigning Bayesian evidence separates candidate models into more and less likely.** (A) Aspects of linear regression model assessed by model selection and model averaging (see Fig 1A). (B) Aspects of mass-action kinetics model / ordinary differential equation assessed by model selection and model averaging. (C) Schematic representation of the equation in (B). (D)-(F). Evidence values (left y-axis) and posterior probability values (right y-axis) from nested sampling, one point per model, ordered from model with greatest evidence to model with least evidence. Models whose evidence value are within $10^{1/2}$ of the greatest evidence value, the "relative likelihood confidence interval," are colored in red. Nested sampling and evidence calculation is performed per dataset. (D) TKO dataset. (E) RPM dataset. (F) SCLC-A cell line dataset. (G) Numbers and percentages of models in the relative likelihood confidence interval, 95% confidence interval, and remaining non-confidence interval models. TKO, $p53^{fl/fl};Rb^{fl/fl};p130^{fl/fl}$ tumors [28]; RPM, $Rb1^{fl/fl};$

$Trp53^{fl/fl}$;Lox-Stop-Lox[LSL]-$Myc^{T58A}$ tumors [36]; SCLC-A cell lines, a subset of SCLC cell lines from the CCLE [54] that we previously assigned as representative of tumors made up largely of the SCLC-A subtype [33].

While a 95% CI is a traditional approach, (Fig 4D–4F, orange) we also calculate a "relative likelihood confidence interval," as discussed in [12] (see Methods). For this relative likelihood CI, we calculate the Bayes Factor (BF) between the highest-scoring model and every other model, using the least strict cutoff of BF > $10^{1/2}$ (as above, and in S1 Text **Note A**). Even with this permissive cutoff, the relative likelihood CI includes only tens of models per dataset, a large decrease from the initial number of candidates (~1% or less, Fig 4G).

Because we assigned all candidate models equal likelihood *a priori*, if post-nested-sampling, no candidate models were informed by the data, each model would have a similar, though non-identical, marginal likelihood. In such a situation, it is likely that approximately 95% of models would fall within the 95% CI. However, our results indicate that 14–26% of candidate models (depending on the dataset) fall within the 95% CI (contain 95% of the probability in the model space). Therefore, we consider that the data used for model fitting has informed our knowledge about the system, because before nested sampling, all models are equally likely.

In summary, we can determine a subset of candidate models that adequately represent the data, conditional on the fitted parameter sets resulting from the model optimization in nested sampling. Investigating these parameter sets can provide more insight into the similarities and differences between candidate models and their fits within and between datasets. Moving beyond the parameter values assigned to each model term, we wanted to investigate how the data available can inform model terms. If data does not inform model terms and variables and the corresponding fitted parameter rates, it indicates that the mechanistic conclusions we desire to draw from this data using mathematical modeling may require additional or different data.

## High-likelihood model topologies are nonoverlapping between datasets

Given our observation that no one candidate model stands out among other models to explain the experimental data, we employed the multimodel inference technique of Bayesian model averaging (BMA). Briefly, the reasoning behind BMA is that a combination of candidate models will perform better in explaining the data than a single model [13]. In BMA, each model is weighted by its posterior model evidence [14] and the model terms within each model receive an averaged likelihood [23] (S1 Text, **Notes A and B**).

To investigate model-averaged parameters, we considered that initiating subtype may affect fitted parameter rates in that, for example, the rate of a phenotypic transition from SCLC-A to SCLC-Y may be different in models where A initiates the tumor vs. a model where Y initiates the tumor. The choice of initiating subtype was not informed by our data (S5 Fig). Had our analysis resulted in a likely initiating subtype (per dataset), we would select that option to constrain initial subtype conditions; since we were left with approximately equivalent probabilities for initiation, we turned to the literature to impose stricter constraints about initial subtype conditions. As mentioned previously, reports link NE SCLC subtypes and long-term tumor propagation [28,39] and, in particular, cells of subtype A [34]. We thus used only candidate models with an initiating subtype of A, with or without other initiating subtypes. Since we required that subtype A be an initiating subtype, model structures that do not include subtype A received zero posterior probability (models 3 and 8–10 in Fig 5A; model topology probabilities without filtering by initiating subtype are shown in S6A Fig).

We perform BMA across all models for each dataset. As shown in Fig 5A, all datasets (TKO, RPM, and SCLC-A cell lines) support both two- and three-subtype topologies. Higher

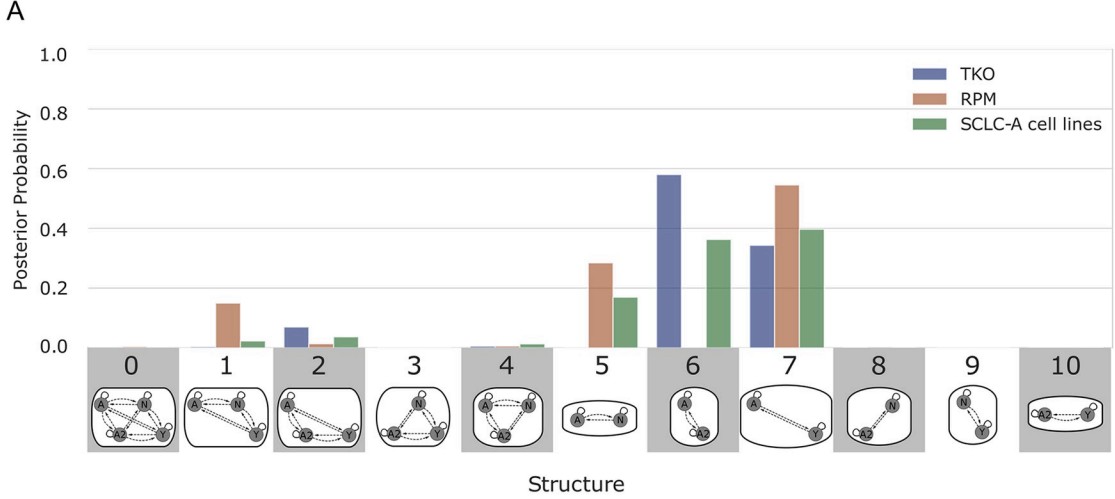

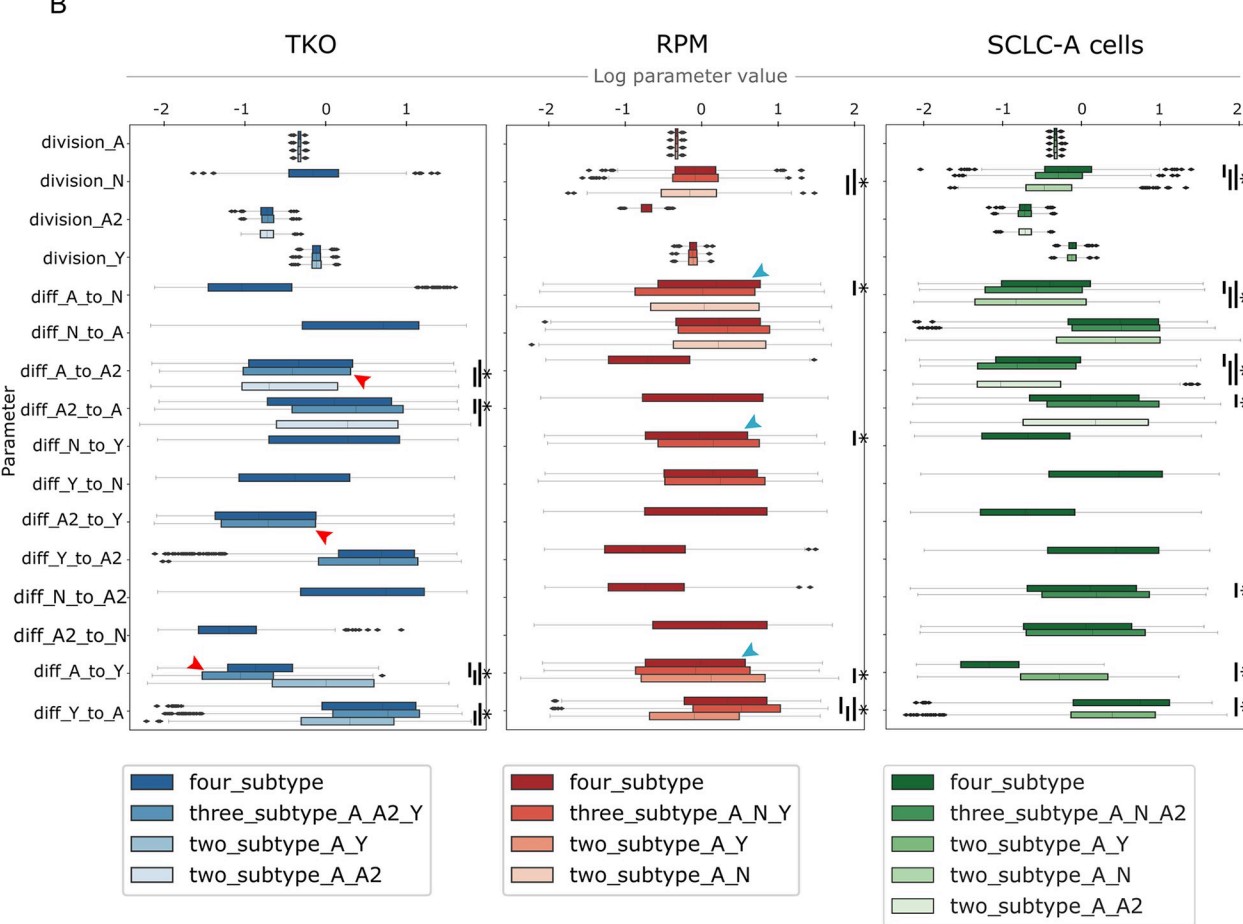

**Fig 5. Likely model topologies vary across datasets; transition rates vary according to subtype presence in similar ways.** (A) Hypothesis assessment of model topologies, per dataset. Probability indicates the result of Bayes theorem using equivalent prior probabilities per topology (e.g., 9% probability that one of the topologies in the x-axis best represents a dataset) and Bayesian evidence values (marginal likelihoods) summed per topology. Model topologies represented by images and corresponding numbers along the x-axis. Posterior probability based on marginal likelihoods of all candidate models that include A as an initiating subtype. (B) Division and phenotypic transition parameters for TKO, RPM, and SCLC-A cell line datasets, comparing between higher-probability topologies (A) and four-subtype topology per dataset. Red arrowheads indicate higher A-to-A2 transition rate in 3-subtype TKO topology (A, A2, Y) compared to A-to-Y and A2-to-Y. Teal arrowheads indicate higher A-to-N transition rate in 4-subtype RPM topology compared to A-to-Y and N-to-Y. TKO, $p53^{fl/fl};Rb^{fl/fl};p130^{fl/fl}$ tumors [28]; RPM, $Rb1^{fl/fl};Trp53^{fl/fl};$Lox-Stop-Lox[LSL]-

*Myc*<sup>T58A</sup> tumors [36]; SCLC-A cell lines, a subset of SCLC cell lines from the CCLE [54] that we previously assigned as representative of tumors made up largely of the SCLC-A subtype [33]. (*) indicates significance between samples from BMA parameter distributions at family-wise error rate (FWER) = 0.01, averaged over ten sampling iterations using one-way ANOVA plus Tukey HSD.

probabilities for two-subtype topologies are expected given that nested sampling prioritizes model simplicity and goodness of fit [60]. Statistically, this result suggests that a two-subtype model could be used to interpret the data reasonably well, but it also shows that topologies comprising three subtypes cannot be excluded. As noted previously, we also withheld prior knowledge from our analysis that tumors with only two subtypes are less likely to explain the data (as new subtype(s) may appear and are thus phenotypically accessible upon perturbation [6,28,33,34,36,38]). Given that our data used (Fig 3) does not include perturbations, it is less surprising that two-subtype models have the highest probabilities. Thus, we consider these two-subtype topologies important, but we place additional importance on three-subtype topologies given our understanding of the SCLC system. Approximately 10% of the probability for the GEMM datasets (TKO and RPM) fall in the three-subtype topology that encompasses the high-probability two-subtype topologies (model 1 for RPM and model 2 for TKO in Fig 5A). For the models fit to SCLC-A cell line data, most of the probability occurs in the topologies with higher probabilities for the GEMM data. This is reasonable, given that SCLC-A cell line data appears as an intermediate between the GEMMs (Fig 3A). However, the SCLC-A cell line data also has probability that falls in the A, N, and A2 topology (model 4 in Fig 5A)—this is the only topology at all likely to represent the SCLC-A cell line data but not at all likely to represent the other two datasets. We interpret the spread of probabilities across multiple topologies, and that most topologies either are probable as representing either TKO or RPM data but not both, to mean that data coverage from these datasets is not sufficient to support one unifying topology. Therefore, each dataset supports a different representation of SCLC tumor growth given its particular (epi)genetic background and environment. This does not mean that a unifying topology or unifying model of SCLC growth cannot exist, but that the biases underlying the experimental data result in different explanations for tumor growth mechanisms.

## All datasets support alteration of phenotypic transition rates in the presence of N or A2 subtypes

After establishing that multiple model topologies can explain tumor growth mechanisms, and given our multimodel inference results from two-, three-, and four-subtype models, we wanted to compare kinetic parameters across models to learn about dynamic variation between model topologies. We aim to make predictions about how the presence of a subtype in the system may affect cell behaviors. We again use BMA to attain this goal, applying the approach to fitted kinetic parameter distributions from nested sampling. In this setting, parameter values from more likely models are assigned higher weights and corresponding parameter distributions are weighted accordingly [14].

While we compare changes in model-averaged parameter rate distributions across model topologies, we do not aim to draw conclusions about specific fitted parameter rates themselves and what a rate may mean for a phenotypic transition in isolation. The nature of the data used for our multimodel inference process cannot provide the information necessary to predict the exact rate of a phenotypic transition, as the small number of datapoints, the steady-state nature of the data, and model topology affect parameter fitting. However, bringing together the data and model topologies to the parameter fitting process can help predict trends in how subtypes, and changes in subtype presence, affect parameter rates in a general sense.

The highest likelihood model topologies (Fig 5A, blue) for the TKO GEMM data, along with the four-subtype topology, are compared in Fig 5B (left). Four model terms have significantly different parameter rates across model topologies, all of which are discussed in S4 Text. Here we highlight differences in the A-to-Y and A-to-A2 transitions across model topologies in the TKO dataset: the A-to-Y transition has a slower rate if A2 is present in the population; Y affects the A-to-A2 transition, increasing its rate. The mechanistic implication of these observations is as such: A2 may represent an intermediate subpopulation in the tumor that is longer-lived, and will only slowly transition to Y. In the topology with A, A2, and Y (Fig 5A, structure 2), the A-to-A2 transition takes up more of the flux in the network than the other hierarchical (NE-to-NonNE directional) transitions (Fig 5B left and S6B Fig, red arrowheads). Additionally, the N-to-Y transition is faster relative to the A2-to-Y transition (S6B Fig, red bar), suggesting that N is a shorter-lived intermediate in the A-to-N-to-Y transition. This result aligns with previous experiments [34] where N was identified as a short-lived state in the A-to-N-to-Y transition. We therefore predict that A2 and N are involved in regulating the relative abundance of, and flux between, A and Y in the tumor.

We also compared the highest likelihood model topologies (Fig 5A, red) for RPM-fitted models, as well as the four-subtype topology (Fig 5B, middle). Five model term parameter rates are significantly different across model topologies, and we highlight again the A-to-Y transition and here the A-to-N and N-to-Y transitions (see S4 Text for discussion of the remaining significantly different parameter rates). The same A-to-Y transition affected in the TKO-fitted models is affected in the RPM model in the same way (reduced rate via an intermediate, in this case N). Here, the A-to-N transition is not affected by Y, but instead increased by the presence of A2; A2 also decreases the N-to-Y transition rate. These similar effects on the A-to-Y transition occur despite the experimental data used for BMA being different. We thus predict that N and A2 are modulating the transition between, and relative abundance of, A and Y. Unlike in the TKO data, when A2 is present in the RPM tumor background, (topology with A, N, A2, Y; Fig 5A, structure 4) the flux through the system spends more time in the N subtype, with more frequent transitions from A to N and less frequent transitions from N to Y (Fig 5B middle and S6B Fig, teal arrowheads). We predict that while N may be a shorter-lived intermediate than A2, A2 regulates the flux from A-to-N-to-Y.

Next, we compared the highest likelihood model topologies (Fig 5A, green) for the SCLC-A cell line data and the four-subtype topology (Fig 5B, right). Seven model term parameter rates are significantly different across model topologies, five of which recapitulate rate alterations based on the presence or absence of different subtypes in TKO or RPM datasets, including the rate alterations discussed above (see S4 Text for more detail).

In summary, BMA enabled us to determine that the A-to-Y transition is regulated in a similar manner for the RPM, TKO, and SCLC-A. Using the higher likelihood model topologies and model-averaged parameter sets, we can infer features of the SCLC tumor generally, despite disparate datasets. Finding the same or similar effects on kinetic parameter rates across independent datasets lends more weight to these predictions about how the N and A2 subtypes may regulate the system flux from A to Y through intermediates and is an advantage of our methodology using Bayes-MMI to work toward a unifying model of SCLC tumor growth based on multiple datasets.

## Model analysis supports a non-hierarchical differentiation scheme among SCLC subtypes

We have considered candidate models (Fig 4), model topologies (Fig 5A), and kinetic parameters (Fig 5B) to explore tumor growth mechanisms in SCLC. There is compelling experimental

evidence for multi-subtype tumor composition, which implies multiple potential growth mechanisms [6,28,34,36,38]. We therefore focused on model topologies 1, 2, and 4, which are three-subtype topologies with detectable probability ($>$ 1%) (Fig 5A), along with the four-subtype model. Using these, we integrate candidate models, topologies, and kinetic parameters, investigating phenotypic transitions between subtypes, whether the presence of certain subtypes affects the behaviors of other subtypes, and if so, which subtypes bring about the effects (Table 3). We conclude by proposing a unifying four-subtype model of tumor growth in SCLC, aiming to represent with one model the varying growth mechanisms accessible across datasets.

We investigate the posterior probabilities, and therefore posterior odds, of each model term (see Methods). Despite different posterior probability values (Fig 6A), the probabilities of model terms across datasets were similar in their trends: across all three-subtype topologies, phenotypic transition probabilities were all more than ½ (Fig 6A, red squares). While some probability values were poorly informed (light red), (probability between ½ and ⅔), more were informed by the data (deep red) (⅔ or more). Conversely, probabilities of Non-NE effects on the growth or transitions were all less than ½ (Fig 6A, blue squares). Some probability values were poorly informed, (light blue) (between ⅓ and ½) and others were informed (deep blue) (⅓ or less) with the addition of data.

Overall, the data suggests that Non-NE effects on transition rates of N-to-Y, or A2-to-Y, are unlikely, (Fig 6B–6D, deep blue) regardless of whether "Non-NE" defines only the Y subtype, or both A2 and Y are Non-NE (Fig 2C). Inter-subtype effects on SCLC phenotypic transition

**Table 3. Probabilities after hypothesis exploration using Bayesian multimodel inference.**

Informed high posterior probabilities
- Simulated tumors appeared more likely to be made up of fewer than four subtypes, indicating the model selection algorithm's preference for parsimony (fewer subtypes to explain the same data).
- Phenotypic transitions A-to-N, N-to-Y, A-to-Y had posterior probabilities between 61% and 75%
  - Posterior odds for these are between ~1.5 and 3.0.
- Transitions are bidirectional: phenotypic transitions N-to-A, A2-to-A, Y-to-N, Y-to-A2, and Y-to-A, had posterior probabilities between 63% and 82%
  - Posterior odds: ~1.5 to 4.5.
- Transitions between N and A2 (N-to-A2, A2-to-N) had posterior probability 69%
  - Posterior odds: ~2

Informed low posterior probabilities
- Low probability of effects that lead to more/quicker phenotypic transitions from NE to Non-NE subtypes, posterior probabilities between 17% and 46% (average 33.4%)
  - Posterior odds: ~0.5
- In SCLC-A cell line datasets, trophic effects, where Non-NE subtypes increase NE division and decrease NE death, had posterior probability 16%
  - Posterior odds: 0.19

Uninformed posterior probabilities
- Initiating / early post-initiation number of subtypes: out of 15 model initiation options (6.67% prior per initiation hypothesis), each probability was between 0.2% and 19.6%
- Phenotypic transitions A-to-A2 and A2-to-Y had posterior probabilities between 53% and 69% (average 58.7%)
  - Posterior odds for these are between 1.13 and 2.2 (average 1.42).
- In TKO and RPM datasets, trophic effects, where Non-NE subtypes increase NE division and decrease NE death, had posterior probabilities between 44% and 45%
  - Posterior odds: 0.79 to 0.82.
- In TKO three-subtype models (only three-subtype model with both types of effects) A2 and Y effects are 54% probable *vs*. Y only effects at 46% probable
  - Posterior odds for A2 and Y effects is 1.17, posterior odds for Y-only effects is 0.85.

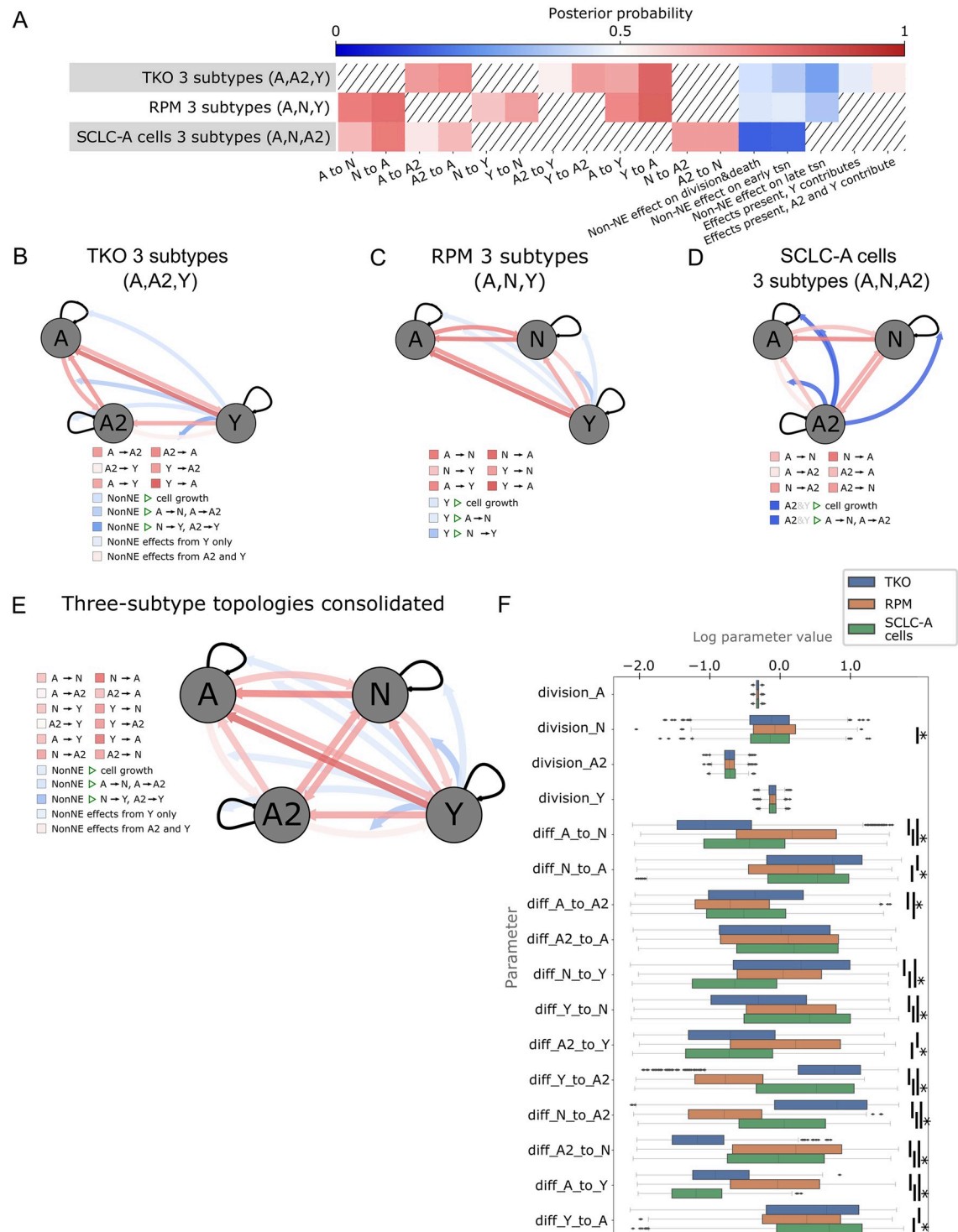

**Fig 6. Across datasets, multimodel inference indicates likely bidirectional phentoypic transitions, suggesting high SCLC phenotypic plasticity.** (A) Heatmap for high probability three-subtype topologies for each dataset (rows), all models initiated by A +/- other subtypes. Color represents the probability of each cellular behavior (column). Since prior probability starts at 0.5 (white), deeper colors indicate a larger deviation from the prior, with red vs blue indicating more likely or less likely, respectively. (B)-(D). Model schematics with each cellular behavior represented by edges coming from or moving toward each cell subtype, (gray circles) growth rates, (self-arrows) or transitions (arrows between gray circles). Edge colors correspond to colors for that behavior in the heatmap in (A). Top-scoring three-state topology for TKO dataset (B), RPM dataset (C), and SCLC-A cell line dataset (D). (E) Schematic of consolidated model behaviors, drawn from each dataset's high-probability three-subtype topology results ((B)-(D)). When multiple

dataset results included different posterior probabilities for a model feature, the one closest to 0.5 was chosen (most conservative). Edge colors correspond to posterior probabilities, with intensity of colors representing information gained from data, as in (A)-(E). (F) Parameter fitting results (part of the nested sampling algorithm) for four-subtype topology models initiated by A +/- other subtypes, across datasets. tsn, transition (e.g., subtype transition). TKO, $p53^{fl/fl};Rb^{fl/fl};p130^{fl/fl}$ tumors [28]; RPM, $Rb1^{fl/fl};Trp53^{fl/fl}$;Lox-Stop-Lox[LSL]-$Myc^{T58A}$ tumors [36]; SCLC-A cell lines, a subset of SCLC cell lines from the CCLE [54] that we previously assigned as representative of tumors made up largely of the SCLC-A subtype [33]. (*) indicates significance between samples from BMA parameter distributions at family-wise error rate (FWER) = 0.01, averaged over ten sampling iterations using one-way ANOVA plus Tukey HSD.

rates have not previously been studied and our analysis predicts that at least effects on "late transition" (Fig 6A), those interactions affecting N-to-Y or A2-to-Y, are unlikely to exist. By contrast, transitions involving A-to-N, N-to-A, A2-to-A, N-to-Y, Y-to-N, A2-to-Y, Y-to-A2, N-to-A2, A2-to-N, A-to-Y, and Y-to-A had posterior probabilities informed by the data (Fig 6B–6D, deeper red). We interpret these results as transitions being likely, i.e., our degree of belief in these transitions has increased. Investigating initiating events via one or multiple cells of origin across the candidate models, we find that from equal prior probabilities of 6.67% per initiating subtype(s) (S2G Fig) the posterior probabilities are not significantly altered, being between 0.2% and 19.6% (S5 Fig). Thus, initiating subtype events were poorly informed by the data. Additionally, analyzing specific model terms, inter-subtype effects on NE subtype growth, inter-subtype effects on transition rates between A and N, or A and A2, and the A-to-A2 transition, were also poorly informed by the data (Fig 6B–6D, light blue, light red).

Finally, to consolidate phenotypic transitions and cell-cell interactions into a unifying mechanism for SCLC tumor growth, we integrated model probabilities from each of the three-subtype topologies for each dataset into one model (Fig 6E). Briefly, phenotypic transition probabilities were chosen from the models least informed by the data in an attempt to make conservative predictions (for more details, see Methods). Model-averaged parameter rates were visually compared (Fig 6F) to ensure that they were within reasonable bounds and that transition rates relate to each other between datasets similarly to our analyses using high-probability topologies (Fig 5B). We chose to consolidate datasets in this way rather than performing multimodel inference on all datasets together: based on our likelihood function, (see Methods) evaluating parameters by comparing simulations to each dataset is equivalent to averaging subtype compositions across datasets and fitting to those proportions. Averaging compositions across datasets leads to a misleading subtype composition that is not representative of SCLC biology, and no information would be gained about any known SCLC system; as such, we consolidated three-topology models as described here instead.

Taken together, these results provide insight not only into what model terms and variables the data is able to inform, but SCLC tumor behavior as well. Knowledge of trophic effects provided by Non-NE cells to the benefit of NE cells was not provided by this particular data; therefore, we cannot use it to understand this behavior. However, we were able to gain knowledge about the likelihood of phenotypic transitions, in fact indicating that nearly all options for phenotypic transitions are likely to exist. We interpret this as high SCLC plasticity, supporting a non-hierarchical differentiation scheme where tumor population equilibrium is achieved through any phenotypic transitions (Fig 6E). It is also clear that consolidating the results across different tumor types is an important step in order to achieve a broader view of the SCLC tumor as a system rather than as one particular experimental model.

## Discussion

The experimental data used for this analysis favors two-subtype topologies as higher-probability candidates. This is not surprising, because nested sampling prioritizes simpler models.

However, studies in SCLC [6,28,34] have shown that phenotypes not measurable in a population initially may emerge upon perturbation, such as drug treatment [40–45] or changes in microenvironment or related factors [34,46]. The data used for our analyses is steady-state data, and we require candidate model simulations to reach steady state as part of the model-fitting process. As such, with this data we are unable to model subtypes emerging upon perturbation in one sample or dataset. Thus, all four subtypes in one tumor type are not captured in our analysis, resulting in the four-subtype topology receiving very low probability. Even so, we consider the four-subtype topology in Fig 6 a potential unifying mechanism, as it enables us to explain the behavior of the three different SCLC genetic backgrounds (TKO GEMM, RPM GEMM, human SCLC-A cell lines) via one model. However, context is necessary in evaluating a potential hypothesis; to predict specific behavior in the TKO GEMM, it is likely optimal to make predictions using the three-subtype model in Fig 6B.

Despite our prior knowledge that more than two subtypes are likely to be accessible, we were interested in the use of Bayes-MMI more generally. We thus aimed to perform a less biased approach, evaluating two-, three-, or four-subtype models. Importantly, having evaluated two-subtype models, we are able to compare trends in parameter rates across models in adding or removing particular subtypes (Fig 5B). While we cannot identify exact parameter rate values for phenotypic transitions, our analysis allows us to make predictions about how the presence of a subtype in the system may affect cell behaviors. The Bayes-MMI workflow enabled us to perform this analysis as well as evaluate the likelihood of each biological process in Table 2 through model-averaged probabilities across candidate models with and without that process (Table 3). Therefore, we highlight that despite our use of steady-state data, we can learn about SCLC tumor growth and draw conclusions about its behavior with accompanying probabilities of *how likely* each conclusion is.

The posterior probabilities for some aspects of our models did not increase or decrease from prior probabilities after incorporation of data, namely cell-cell interactions: we consider these biological processes "uninformed" by the data (Table 3). We hypothesize that to evaluate models with such interactions, data measuring transient dynamics, such as found in [62] will be necessary to perform model selection. With this, we may be able to determine the probability of cell-cell interactions related to growth, as well as the origin of these interactions (whether from a "NonNE compartment" represented by A2 and Y, or by Y alone). We also note that the invasive or metastatic potential of the SCLC tumor is known to be increased by Non-NE subtypes [29,63].

Future work will better evaluate the optimal SCLC model topology(ies), cell-cell interaction effects, cell-cell interaction origins, and the physiologic context in which tumors may exist. This can be done by incorporating time-course data, perturbation data, and data from tumors both *in situ* and during invasion. Perturbation and likely environmental data (local vs invasive tumor) will enable the measurement of, and thus fitting to, more subtypes in our data. Time-course data will enable the assessment of cell-cell interactions. Most ideally, experiments measuring subtypes, initial conditions, and tumor population growth and change over time, such as the model selection analysis done in [62], could be designed with our multimodel inference workflow in mind. This would then combine experimental and computational perspectives, to enable biological hypotheses to be most optimally assessed.

The results presented here provide strong evidence for phenotypic plasticity in SCLC tumors, based on the higher likelihood for most phenotypic transitions tested, regardless of differentiation hierarchy. We cannot claim that the inferred rate values of phenotypic transitions are precise, an endeavor limited by the data at hand, but we can evaluate the likelihood of biological hypotheses represented in our models, with focus on the likelihood of phenotypic transitions. The general question of whether phenotypic transition hypotheses are more or less

likely in SCLC tumor growth, is answered using all possible initiating subtype combinations in each condition, and we find that the data supports phenotypic transitions over lack of cellular transitions. With a more plastic and less stem-cell based phenotypic equilibrium, instead of rare remaining stem-like populations leading to regenerate a tumor after treatment, we hypothesize that any SCLC subtype that remains post-treatment can lead to tumor regeneration and subsequent treatment resistance, patient morbidity and mortality. In considering hierarchical phenotypic heterogeneity vs. phenotypic plasticity, we would propose evaluating an SCLC population for our highest-probability non-hierarchical transition, the Y-to-A transition: sorting a population for SCLC-Y cells and evaluating over time for the appearance of SCLC-A cells would test our prediction that the Y-to-A transition exists. In fact, such plasticity in general, and the Y-to-A transition specifically, has recently been experimentally studied in SCLC, where Gopal and colleagues measured the Y-to-A transition in untreated *ex vivo* PDX cells, among other nonhierarchical transitions [64]. In keeping with our predictions that plasticity is highly likely, Gopal and colleagues found the Y-to-A and other nonhierarchical transitions. It continues to be of particular interest to compare phenotypic plasticity and the prevalence of non-hierarchical transitions in treated *vs.* untreated tumor samples, as treatment is likely to alter the mechanisms by which tumor population equilibrium is maintained. Time-course experiments with surface marker labeling or live-reporter imaging can resolve and provide confirmation for bidirectional phenotypic transitions in the treated or untreated context, which are crucial to understand in order to battle SCLC treatment resistance.

We believe a shift from information theoretic multimodel inference toward a Bayesian approach, in the context of model averaging especially, will benefit modeling in systems biology. The methodology employed herein incorporates model selection and model averaging into a multimodel inference framework, followed by Bayesian analysis to identify not only whether a hypothesis investigated via mechanistic modeling is or is not likely, but *how* likely (and thus how informed by the data) that hypothesis is. Understanding which hypotheses are informed by the data is especially important given variability between data in investigations of the same systems, such as a particular tumor type. It is difficult to attain a consensus model since investigators use varying experimental models within the same physiologic or disease process and thus may draw nonoverlapping conclusions, building parts of a picture but not a whole. Striving for the whole picture, via principled statistical analysis, to be followed by experiments based on informed model predictions, will advance cancer research and lead to better treatments.

## Methods

### CIBERSORT deconvolution of RNA sequencing data

Data from two GEMM models provide multiple replicates of tumors from two genetic backgrounds: one from $p53^{\text{fl/fl}}$;$Rb^{\text{fl/fl}}$;$p130^{\text{fl/fl}}$ (triple-knockout, or TKO) GEMM tumors [28], and another from $Rb1^{\text{fl/fl}}$;$Trp53^{\text{fl/fl}}$;Lox-Stop-Lox[LSL]-$Myc^{\text{T58A}}$ (RPM) GEMM tumors [36]. We also used publicly available SCLC cell line data. Having been originally derived from human tumors, each cell line has a different genetic background, and therefore we have only one (genetically identical) replicate per cell line sequencing event. To approximate genetic similarity between cell lines, and thus approximate multiple replicates, we expect that cell lines exhibiting similar steady state composition will be more genetically similar than those whose steady state compositions differ. Previously, in [33], we both clustered publicly available SCLC cell line data into clusters that align with the different SCLC subtypes and used CIBERSORT to deconvolute the proportions of cell line data and tumor samples into SCLC subtypes from their RNA sequencing signatures. Results used for this publication can be found as S1 File.

## Population dynamics modeling in PySB

A population dynamics model represents the abundance of species over time, whether increase or decrease due to birth/growth or death. We use ordinary differential equation (ODE) models coded via PySB to generate population dynamics models [65]. PySB is a rule-based modeling language, where one will encode

$$A \xrightarrow{k} 2A, \; k = 0.469 \qquad (1)$$

to indicate that A doubles at a rate of 0.469 doublings per day. PySB then generates the ODEs from encoded rules, such as the following related to the single rule in Eq (1):

$$\frac{dA}{dt} = k[A] \qquad (2)$$

Inter-subtype effects are represented by the increase or decrease of the rate of affected reaction. For example, the above division rule has a baseline rate of 0.469 doublings per day, but in the presence of an effector subtype the division rule will have a rate of 0.469*1.05 = 0.493 doublings per day. In this case the effector subtype has increased the division rate by 5%. Thus the rule-based representation is

$$A + Y \xrightarrow{k*} 2A + Y, k^* = 0.493 \qquad (3)$$

$$A \xrightarrow{k} 2A, \; k = 0.469$$

to indicate that while A doubles at a rate of 0.469 per day, it doubles at a rate of 0.493 doublings per day in the presence of Y. PySB will generate this ODE from the encoded rules:

$$\frac{dA}{dt} = k[A] + k^*[Y][A] = (k + k^*[Y])[A] \qquad (4)$$

To simulate the passage of time, the speed at which the division/death/transition reaction occurs–its rate, $k$, of cells per unit time–must be assigned as in the equations above. While a literature search reveals approximate rates of division and death among different SCLC subtypes (S1 Table), each of these are in a different context than the system we model here—for example, division rates for the A subtype are measured *in vitro* in the presence of only that one subtype, whereas our population dynamics model is meant to simulate this subtype in the presence of others as well as *in vivo* in a mouse tumor. Therefore, we use the rates in the literature as our prior expectations for division and death, that is, we use these values as approximate starting values for these parameters during the estimation process. Other rates, such as those indicating the speed of transitions between subtypes, or any rates including the effects of Non-NE subtypes, have not previously been noted in the literature and we used much wider ranges for each as our prior expectations. Rate prior expectations (S3 Fig) are then provided to the Multinest algorithm to perform nested sampling.

## Multiple hypothesis generation via HypBuilder

Because we perform model selection, we use 5,891 ODE models coded via rule-based modeling in PySB. Each model is generated to include or exclude from 44 reaction rules. There are eight rules that represent division and death for each subtype, and with the potential for three different inter-subtype effects (including none) to have an impact on division or death, each division and death reaction has 3 options, leading to 24 potential rules relating to division/death in total. There are four rules that represent hierarchical phenotypic transitions, which likewise

have three potential inter-subtype effects, for 12 rules in total representing hierarchical phenotypic transitions. There are eight rules related to non-hierarchical phenotypic transitions, for 20 total potential phenotypic transition rules out of the 44 rule options.

We use HypBuilder (https://github.com/LoLab-VU/HypBuilder) to automatically generate the 5,891 PySB models that we would otherwise have to code by hand. HypBuilder is software for the automatic generation user-specified collections of mechanistic rule-based models in the PySB format. The input CSV file contains a global list of all possible model components, and reactions, as well as any instructions regarding model creation. The instructions dictate which subsets of model components and reactions will be combinatorially enumerated to create the collection of models. The reactions are parsed via HypBuilder's molecular interaction library, a library of defined reaction rule sets that is outfitted with common PySB interactions and is customizable to include more interactions should the user need them. Once parsed and enumerated each combination of rules is exported as an executable model via PySB.

The instructions for model construction used in this work direct HypBuilder to use a "list" method to enumerate all candidate models of interest using prior knowledge of likely combinations of model variables (see https://github.com/LoLab-MSM/Bayes-MMI for code used to enumerate candidate models and create the list for HypBuilder).

If the candidate model set contains every relevant biologically plausible possibility, we can consider the entire set of models as representative of 100% of the probability that one of the candidate models explains, or provides the mathematical basis underlying, the data. This is an assumption that cannot truly be met, and most model selection literature acknowledges that one cannot find the "true" model [12,16]. However, prior knowledge enables us to determine that all 5,891 models represent all possibilities with regard to outstanding SCLC hypotheses to the best of our ability.

We visualize the prior expectations for the 44 rate parameters as a probabilistic distribution per parameter (prior marginal distribution) (S3 Fig). Correspondingly, a probabilistic representation of best-fitting rates for each model is returned by the Multinest algorithm (posterior marginal distribution) (e.g., Figs 5B and 6E and S6B Fig).

## Parameter estimation and evidence calculation by nested sampling

As noted in Eqs (1) and (3), rate parameters must be set in order to run simulations of a mathematical model. Parameter estimation is the process of determining optimal rates that result in a model simulation recapitulating the data it is meant to represent. Multiple methods exist for parameter fitting or model optimization, [66,67] with Bayesian methods utilizing a prior rate parameter distribution, $P(\boldsymbol{\theta})$, where $\boldsymbol{\theta}$ represents the set of $n$ parameters $\{\theta_1, \theta_2, \ldots, \theta_n\}$, and a likelihood function to assess a parameter set

$$L(D|\boldsymbol{\theta}_i) = P(D|\boldsymbol{\theta}_i), \tag{5}$$

Where $\boldsymbol{\theta}_i$ is the $i^{\text{th}}$ parameter set and $D$ represents the data being used for fitting. Parameter set $\boldsymbol{\theta}_i$ is scored via the likelihood function $L(D|\boldsymbol{\theta}_i)$ and optimization continues, moving toward better-scoring parameter sets until an optimal score is reached.

With a prior probability, $P(\boldsymbol{\theta})$, and a likelihood (Eq 5) the posterior probability can be calculated via Bayes' Theorem,

$$P(\boldsymbol{\theta}_i|D) = \frac{P(D|\boldsymbol{\theta}_i)P(\boldsymbol{\theta}_i)}{\int P(D|\boldsymbol{\theta})P(\boldsymbol{\theta})d\boldsymbol{\theta}} \tag{6}$$

The denominator of Bayes' Theorem represents the likelihood integrated over all parameter sets, called the marginal likelihood or model evidence. Nested sampling computes this value (Skilling, 2004).

To perform nested sampling, we utilize the Multinest algorithm [19–21]. Multinest samples multi-dimensional parameter space, bounding its search by parameter values along each axis in each of the multiple dimensions based on prior expectation of parameters, $P(\theta)$ input by the user. It removes the lowest-probability parameter set and chooses a new one from within the bounded parameter space, subsequently re-drawing the search space with the bounds incorporating the new parameter set. This continues until all parameter sets representing the bounds of the search space have approximately equal probability, and the algorithm estimates that the remaining probability of parameter sets within the bounds is less than a user-defined tolerance. Each parameter set is evaluated based on a user-defined likelihood function (Eq 5). Finally, the likelihood values that correspond to each sampled parameter set are arranged in the order they were replaced, and the integral over these is taken to approximate the integral over all possible models, that is, the marginal likelihood or Bayesian model evidence.

## Nested sampling marginal likelihood calculation for comparison to AIC analysis

For this analysis we used simulated data generated via code provided by Galipaud and colleagues [15] (see S1 Text **Note A.7**). This data includes values for one response variable $y$ and four predictor variables $x_{1-4}$, so we aimed to use nested sampling to determine coefficients for the linear regression model that minimized the mathematical distance between model simulated $y$ values ($y_{sim}$) and "ground truth" (simulated data) $y$ values $y_{data}$. Here,

$$y_{sim} = \left(\beta_0 + x_{1_{data}} * \beta_1 + x_{2_{data}} * \beta_2 + x_{3_{data}} * \beta_3 + x_{4_{data}} * \beta_4\right), \tag{7}$$

where each $\beta_i$ scalar in Eq (7) is equivalent to a parameter $\theta_i$ (as in (Eq 5)) and represents the value in each $i^{th}$ dimension of parameter space that Multinest searches. For simulating $y$ values $y_{sim}$ during Mulitnest's parameter search, each predictor variable array $x_{i\_data}$ and the response variable array $y_{data}$ values in Eq (7) are from the simulated data (snippet shown in S1 Text **Note A.7**). For prior distributions of each parameter/regression coefficient $\beta_i$, we used uniform prior distributions from 0 to 10 for $\beta_{1-4}$ and a uniform prior distribution -10 to +10 for the intercept $\beta_0$. For the model selection problem, if a predictor variable $x_i$ was not included in the model, its regression coefficient $\beta_i$ was set to zero.

Sampling regression coefficient values across the parameter space and using them and $x_{i\_data}$ values to calculate $y_{sim}$, we then calculate distance between simulation and data via least squares estimation with

$$\frac{(y_{data} - y_{sim})^2}{\sigma^2_{y_{data}}} \tag{8}$$

Using the Multinest algorithm results, the minimal distance, which, assuming normally-distributed error, corresponds to maximum log-likelihood, can be determined out of all regression coefficient sets [68]. Multinest then returns the marginal likelihood (Bayesian evidence) for each candidate model. We used the Multinest-returned "vanilla" nested sampling evidence value [21].

### Nested sampling marginal likelihood calculation for SCLC candidate models and SCLC datasets

For this analysis we used data as described above in "CIBERSORT deconvolution of RNA sequencing data", representing each SCLC dataset's tumor steady-state proportions probabilistically, with a Beta distribution (bounded by zero and one). We selected a Beta distribution as it is a suitable model to capture the behavior of proportions, or relative abundances, [69–71] and we are here fitting to percentage (proportional) data (Fig 3). The Beta distribution recognizes the relationship between mean and variance that is likely to occur with proportions, and a mean close to zero or one generally has a smaller variance than proportions with a mean further from zero or one [70]. In this way, means close to zero and one are likely to have small enough variances for the distribution to be bounded by zero and one without an abrupt cutoff, as would be necessary had we used a normal distribution with a mean near zero or one.

We use the means and variances of previously published sample replicate subtype proportions [33] as our data. For each dataset, then for each subtype in Fig 3A, the mean and variance of the proportion of that subtype across samples is used.

For our likelihood function, we simulate the tumor for 60 time steps (representing days) and require that 1) at 60 days, the total tumor is more than 100 cells and less than too many cells for the ODE solver to handle (in this case the solver returns NaN), and 2) the last 7.5% (~4.5 days) of the simulation, each subtype proportion has reached steady-state, i.e. each subtype's proportion trajectory has a derivative $< 0.0001$ and $> -0.0001$. We then compare each subtype proportion at steady-state in the dataset of interest ($D$) to each simulated steady-state subtype proportion ($\theta$), thus evaluating $P(D|\theta_i)$ (Eq 5). We set the simulated proportion of a subtype (one value per subtype, given that the ODE model is deterministic) as the mean $\mu_{s_{sim}}$, with the variance of the data (Fig 3 and S1 File) as the variance $\sigma^2_{s_{data}}$ [17,72]. Then we used these values to calculate a Beta distribution, that is calculating $\alpha$ and $\beta$ using the mean $\mu$ and variance $\sigma^2$ of each dataset [73]:

$$v = \alpha + \beta = \frac{\mu(1-\mu)}{\sigma^2} - 1, \; where \; v = (\alpha + \beta) > 0 \; (meaning \; \sigma^2 < \mu(1-\mu)) \qquad (9)$$

$$\alpha = \mu v = \mu\left(\frac{\mu(1-\mu)}{\sigma^2} - 1\right), \; if \; \sigma^2 < \mu(1-\mu) \qquad (10)$$

$$\beta = (1-\mu)v = (1-\mu)\left(\frac{\mu(1-\mu)}{\sigma^2} - 1\right), \; if \; \sigma^2 < \mu(1-\mu) \qquad (11)$$

We then calculated the log likelihood of the mean of the data $x_s$ using the constructed Beta reference distribution,

$$\sum_{s}^{S-1} LL \, ,$$

where

$$LL = \begin{cases} log\left(\dfrac{x_{s_{data}}^{\alpha_{s_{sim}}-1}\left(1 - x_{s_{data}}\right)^{\beta_{s_{sim}}-1}}{B(\alpha, \beta)}\right) & if \; s \in simulated \; model \; topology \\[2em] log\left(\lambda e^{-\lambda x_{s_{data}}}\right), \; where \; \lambda = \dfrac{1}{\sigma_{s_{data}}}, \; otherwise \end{cases} \qquad (12)$$

where $S$ is the set of subtypes, $\alpha_{s_{sim}}$ and $\beta_{s_{sim}}$ indicate the calculation of $\alpha$ (Eq 10) and $\beta$ (Eq 11) as discussed above, $x_{s_{data}}$ is the mean proportion of subtype $s$ in the dataset, $B(\alpha,\beta)$ is $\frac{\Gamma(\alpha)\Gamma(\beta)}{\Gamma(\alpha+\beta)}$ and $\Gamma$ is the Gamma function, and $\sigma_{s_{data}}$ is the standard deviation of the data. Using the exponential function $(log(\lambda e^{-\lambda x_{s_{data}}}))$ as part of the log-likelihood (Eq 12) enabled us to calculate a likelihood value for subtypes not present in a model's topology, which should be a poor log-likelihood if the subtype has a high proportion in the data but was not included in the model topology, or a better log-likelihood if the subtype has a low proportion in the data but was not included in the model topology (and therefore potentially contributing to overfitting). The Python module scipy.stats was used to calculate the Beta log likelihood (Eq 12, top) and the exponential log likelihood (Eq 12, bottom). A simulation would not be scored (return *NaN* and thus be thrown out by the Multinest fitting algorithm) if the tumor subtype proportions did not reach steady state (calculated by whether a proportion timecourse had a slope of zero for the last 7.5% of the simulation). We used the Multinest-returned importance nested sampling evidence value, because multiple importance nested sampling runs (multiple 'replicates') for the same candidate model and prior parameters returned more consistent evidence values than 'vanilla' nested sampling [21].

Multinest is run per model per dataset, which equates to performing 5,891 mechanistic interpretations, 3 times each. CPU time for one model fitting was on average 19 hours (~0.80 days), with a range of 5 minutes to 28 days. If Multinest had not reached its stopping point by 28 days, we assumed that all regions of parameter space were similarly unlikely and that further running of the algorithm would only continue to refine the search of the unlikely space; models with this difficulty are very likely to have low marginal likelihood due to the unlikeliness of the parameter space. We do not include these incompletely-searched models in our multimodel inference analyses (Figs 4–6) and we confirmed that all models that reached 28 days of CPU time without reaching the Multinest stopping point have a low evidence value at the time they were terminated.

## Calculating Akaike Information Criterion (AIC) and corrected AIC (AICc)

The AIC balances the maximum log-likelihood score from parameter fitting with the number of parameters $\boldsymbol{\theta}$ in the candidate model; penalizing by number of parameters is meant to reduce bias (see S1 Text **Note A.3** for more detail). The AIC is denoted by

$$\text{AIC} = -2\ln(L(\boldsymbol{\theta}_{best}|D)) + 2K \tag{13}$$

where $L(\boldsymbol{\theta}_{best}|D)$ is the likelihood of the best-fitting parameter set and $K$ is the number of parameters in the model.

The corrected AIC, AICc, is used to account for small sample sizes. We use this for calculating information criterion for models in the simple example of [15]:

$$AIC_c = AIC + \frac{2K^2 + 2K}{n - K - 1} \tag{14}$$

Where *AIC* and $K$ are as used in Eq (13), and $n$ is the sample size of the data.

We chose to calculate AIC (rather than AICc) for the comparison of AIC, BIC, and marginal likelihood for our SCLC data (S1 Fig). This is due to the fact that Eq (14) was derived for a linear regression model with normally distributed errors, which is the case for the dataset in [15], but not for our SCLC multimodel inference analysis [74].

## Calculating Akaike weights (here noted as AICc weights) and sums of AICc weights (SW)

Model AICc scores are first scaled with respect to the minimum AICc value [12,16], resulting in AICc differences,

$$\Delta_k = AICc_k - \min(AICc) \tag{15}$$

Where $min(AICc)$ is the AICc value of the lowest-scoring (best) model candidate, and delta_k is the AICc difference for model k compared to the model with the lowest AICc. Using AICc differences, AICc weights are calculated,

$$w_k = \frac{e^{-\Delta_k/2}}{\sum_{r=1}^{R} e^{-\Delta_r/2}} \tag{16}$$

where $R$ is the number of models in the candidate set. AICc weights represent relative likelihoods of the models given the data, and are interpreted as probabilities [12], which can be seen in Fig 1C, red points.

$$SW_{H_i} = \sum_{r=1}^{R}\{w_r \; if \; M_r \in H_i\} \tag{17}$$

$SW_{H_i}$ is the sum of AICc weights for hypothesis $H_i$, which can represent a model term, but in our example represents the inclusion of a response variable $x_{1-4}$ in the optimal model (Fig 1). Here, the AICc weight $w_r$ for model $M_r$ is summed only if model $M_r$ falls under the hypothesis $H_i$—in the case of Fig 1, if $M_r$ includes the model variable for which the SW is being calculated.

## Candidate model prior and posterior probabilities and confidence interval calculation

Each candidate model is considered equally likely prior to fitting by Multinest. That is, every candidate model has an equal prior probability of being the optimal model to represent the underlying SCLC tumor system,

$$P(M_k) = \frac{1}{|M|}, \tag{18}$$

Where $M$ is the set of all candidate models. With the model evidence, or marginal likelihood, $P(D|M_k)$ estimated by Multinest, [19–21] the posterior probability per model can be calculated as

$$P(M_k|D) = \frac{P(D|M_k)P(M_k)}{\sum_j P(D|M_j)P(M_j)} \tag{19}$$

With a posterior probability per model, (Fig 4) we calculate a 95% confidence interval. This is accomplished by summing decreasing model posterior probabilities until the sum is 0.95, then considering those models as our 95% CI [12] (Fig 4, orange). Using this confidence interval results in ~1000 models per dataset, a considerable decrease from the initial 5,891. This is a more traditional approach to determining a confidence set of models.

We also took an approach discussed in [12]. In this approach, a CI is informed by use of the Bayes Factor between the highest-scoring model and consecutively decreasing scoring models, until the Bayes Factor is larger than a particular cutoff. The models in this CI would be those models I for which $\frac{P(M_{highest})}{P(M_i)} > cutoff$. Burnham and Anderson denote such a method as a

"relative likelihood confidence interval" and discuss its support by statistical theory, noting that it is uncommonly found in the model selection literature [12]. We used a cutoff of $10^{1/2}$, the lowest Bayes Factor at which a difference may be determined [61]. Even with this permissive cutoff, the relative likelihood CI includes only tens of models, an even greater decrease from the initial number of candidates.

## Prior and posterior probabilities per hypothesis being investigated

Each hypothesis has an assigned prior probability based on our prior expectations. For all hypotheses, we took an approach where we considered each hypothesis as equally likely compared to competing hypotheses. For the inclusion of most variables (Fig 1) and model terms (Fig 5), this was a prior probability of 0.5 or 50%, where it is 50% likely the model term is part of a model that is the best representation of the tumor system, and 50% likely that same term is not part of that model. For the inclusion of effects in the candidate models, the prior probability for a given effect is 33%, where it is equally likely that an effect is generated by Y, generated by A2 and Y, or that no effect is present. The comparison between effect types (including none) is included in S2–S4 Tables, while the comparison of any effect at all *vs.* no effect (50% *vs.* 50%) is included in the main text.

For the model topology analysis, we considered it equally likely that any model topology could best represent the tumor system that generated each dataset, and with 11 possible model topologies this resulted in a 9% prior probability per model topology (Fig 5A). For model initiating subtype hypotheses, (S2G Fig) with 15 potential combinations of initiating subtypes, each initiating subtype combination has a 6.67% prior probability.

Each candidate model can then be assigned a prior probability conditional on the hypothesis being considered, $P(H_i)$, where $M_k$ is the $k^{th}$ candidate model and $H_i$ is the hypothesis being considered. The calculation of $P(H_i)$ is based on the number of candidate models that fall under the hypothesis being considered,

$$P(M_k|H_i) = \frac{P(H_i)}{\left|\left\{M_j \in M, H_i\right\}\right|} \tag{20}$$

where $\{M_j \subset M, H_i\}$ is the set of all models assigned to $H_i$. For example, if $H_i$ is the hypothesis that the model term "A to Y transition" is part of the model that would best represent the SCLC tumor system, then $\{M_j \subset M, H_i\}$ is the set of all candidate models that include the "A to Y transition" model term.

Using this prior probability, the posterior probability for an individual model, conditional on the hypothesis being considered, can be calculated as

$$P(M_k|D, H_i) = \frac{P(D|M_k, H_i)P(M_k|H_i)}{\sum_j P\left(D|M_j, H_i\right)P(M_k|H_i)} \tag{21}$$

Where $P(D|M_k,H_i)$ is the Bayesian model evidence (marginal likelihood) for $Model_k$.

The posterior probability for an individual model $k$ under hypothesis $H_i$, $P(M_k|D,H_i)$, is not directly used, as the posterior probability of $H_i$ itself, $P(H_i|D)$ is of principal interest. Under Bayes' Theorem,

$$P(H_i|D) = \frac{P(D|H_i)P(H_i)}{\sum_j P\left(D|H_j\right)P\left(H_j\right)} \tag{22}$$

Where $P(D|H_i)$ is the marginal likelihood of $H_i$ over all models to which it applies, $\{M_j \subset M,$

$H_i$}. According to [61], this can be calculated as

$$P(D|H_i) = \Sigma_k P(D|M_k, H_i) P(M_k|H_i) \tag{23}$$

with a summation instead of an integral because each model has a discrete prior probability as calculated in Eq (20).

Using the results of Eq (23) in Eq (22), we then calculate the posterior probability for each hypothesis, pictured in Fig 6A–6D and noted in Table 3. In this way, we can use Bayesian calculation rather than sums of AICc weights [18,15] (the direct comparison of these in our example is shown in Fig 1C; for additional details see S1 Text **Note A**) to determine the posterior probability of each model term. This also enables us to avoid bias in considering models with and without certain model term, if an uneven number of candidate models contain a model term *vs*. do not contain the term (see S1 Text **Note B**) [75].

### Posterior odds per hypothesis being investigated

All model terms and variables begin with a prior probability of 0.5. With equal prior probabilities across all model term hypotheses, the posterior odds represented by $\frac{posterior\ probability}{(1 - posterior\ probability)}$ is equivalent to the Bayes Factor. Therefore, calculation of the posterior odds and the Bayes Factors for each model term are equivalent.

A posterior probability of model term inclusion of 0.75 or more, or probability of 0.25 or less, would be considered substantial evidence for inclusion or exclusion of that term, respectively [61]. Given the nature of the posterior odds, where a value of 2 indicates that one hypothesis is twice as likely to be true as the other, we also consider posterior probabilities of 0.667 or more, or 0.333 or less, to be notable evidence for inclusion or exclusion of the model term considered. We consider probabilities between 0.333 and 0.667 to not have been significantly informed by the data.

### Bayesian model averaging of parameter sets

Since Multinest returns multiple best-fitting parameter sets, each parameter in a model has a frequency distribution representing the values it takes on over these parameter sets. We thus consider each parameter using a probabilistic representation, per model (posterior marginal distribution) (Figs 5B and 6E and S6B Fig). Since each candidate model is assigned a posterior probability as in Eq (11), all best-fitting parameter sets for that model can be assigned the same posterior probability. The frequency distribution of one parameter's values across a model's best-fitting parameter sets are thus weighted by its model's posterior probability. Then, the frequency distributions of weighted parameter values per model can be combined, representing the distribution of potential values of a particular parameter, weighted by model posterior probabilities. This way, parameter values in the distribution that come from models with a higher posterior probability (thus higher model evidence) will have more of an effect on the probabilistic representation, since they represent more likely values for the parameter.

To assemble representative fitted parameter sets for each candidate model, we used the first 1000 parameter sets from the Multinest equally weighted posterior samples per model. With up to 44 parameters and up to 5,891 models, the collection has 44 parameter columns and up to 5,891,000 rows representing a parameter vector. The collections were made per dataset.

### Comparing parameter distributions

As above, each kinetic parameter has a frequency distribution representing 1000 fitted values per candidate model, meaning up to 5,891,000 fitted values across all models (weighted using

Bayesian model averaging, as above). To compare parameter rates across models in the same dataset but with different topologies, we grouped each parameter according to the model topology from which it came. We then sampled 1000 values from the BMA-weighted distribution per kinetic parameter across all models of the same topology. We performed ANOVA followed by Tukey HSD at family-wise error rate (FWER) of 0.01, using the Python module statsmodels. Below an FWER of 0.01, we considered the sampled parameters significantly different across models. We then repeated the sampling, ANOVA, and Tukey HSD for a total of 10 iterations. We then averaged across determinations of significant/non-significant and if a parameter comparison across model topologies was significantly different more often than it was not different, we considered the parameter rates to be different comparing model topologies. The same methodology was used to compare parameter rates across different datasets.

## Generating a consolidated model of the SCLC tumor

A hypothesis (model term) whose posterior probability is further from its prior probability indicates more information gained during the nested sampling process—more knowledge provided by the data. Conversely, a posterior probability similar to the corresponding prior probability indicates that the data did not inform our prior knowledge.

To unify the varying models into one view of SCLC biology, we brought together model probabilities from each three-subtype topology per dataset (Fig 6E). To bring together the results for each three-subtype topology results in the investigation of what appears as a four-subtype topology. In fact, if we are to envision one model that can represent one system that generated all three datasets, it would need to include all four subtypes. We consider this a reasonable practice in that all transition posterior probabilities in the three-topology subtypes either were little informed by the data or had a value indicating that transitions are likely; in addition, all Non-NE effects were either little informed by the data or had a value indicating that these effects are unlikely. Posterior probabilities were not the same between three-subtype topologies, but these trends of likely or unlikely model features generally agreed.

When consolidating models in this way, if model terms were part of multiple topologies (e.g., the A-to-N transition is part of the A, N, and Y topology, best representing the RPM dataset, and the A, N, and A2 topology, best representing the SCLC-A cell line dataset) we took the posterior probability of the model feature closer to 0.5. For example, the posterior probability for the A to N transition in the RPM dataset is 0.709 and the posterior probability for this same transition in the SCLC-A dataset is 0.626. Therefore, in the four-subtype consolidated representation, the posterior probability for the A to N transition is 0.626. This is the most conservative way to represent the knowledge gained by the data from the perspective of the entire SCLC system, allowing for the most uncertainty to remain. We consider this practice as avoiding claiming more certainty about model features than the data may provide.

## Supporting information

**S1 Text. Didactic example contrasting Akaike Information Criterion and Bayesian posterior probability.** Note A. Contrasting AIC vs posterior probability calculated by Bayes-MMI for model selection and multi-model inferenceA.1. Using multiple models to evaluate how well a variable informs the observed data: an example. A.2. Marginal likelihood or "evidence" is calculated using model optimization followed by Bayes' Theorem. A.3. AIC is calculated as an estimate of the Kullback-Liebler divergence. A.4. Notable differences between AIC and Bayesian evidence / posterior probability. A.5. Model selection allows us to evaluate which variables or terms have the largest effect on observed data. A.6. Model averaging uses model selection outcomes from all models to demonstrate how the observed data informed the model

variables or terms that represent our hypotheses. A.7. Advantages of Bayes-MMI over AIC for model selection and model averaging: continuing example. Note B. Sums of AIC weights (SW) and posterior probability on a subset of the candidate models. Note C. Bayesian Information Criterion. Table A Summary of nested sampling model selection results on the simulated dataset and model selection problem in Galipaud *et al.*, 2014, ranked by AICc. Table B Summary of nested sampling model selection results on the simulated dataset and model selection problem in Galipaud *et al.*, 2014, ranked by posterior probability. Table C SW and posterior probability calculations for each model variable in both full candidate set and partial candidate set examples. Table D Summary of AICc and nested sampling model selection results using a partial candidate set. Table E Summary of nested sampling model selection results on the simulated dataset and model selection problem in Galipaud *et al.*, 2014, ranked by BIC-estimated posterior probability. Table F Comparing BIC-estimated probability and marginal likelihood-calculated probability for each model variable in full and partial candidate sets.
(DOCX)

**S2 Text. Population dynamics modeling of small cell lung cancer.** Note A. Population dynamics modeling and inter-subtype effects. Note B. Ordinary differential equations representing each SCLC subtype in the population dynamics models.
(DOCX)

**S3 Text. Simulations using best-fitted parameters, as opposed to randomly-selected parameters from the prior distributions, replicate subtype proportions at steady state.**
(DOCX)

**S4 Text. All datasets support alteration of phenotypic transition rates in the presence of N or A2 subtypes.**
(DOCX)

**S1 Table. Existing data pertaining to SCLC intratumoral heterogeneity and communication used for rate parameter priors.**
(DOCX)

**S2 Table. Model term posterior probabilities after hypothesis exploration, TKO high-probability 3-subtype topology.**
(DOCX)

**S3 Table. Model term posterior probabilities after hypothesis exploration, RPM high-probability 3-subtype topology.**
(DOCX)

**S4 Table. Model term posterior probabilities after hypothesis exploration, SCLC-A cell line data high-prob. 3-subtype topology.**
(DOCX)

**S1 File. CIBERSORT-deconvoluted data from three SCLC datasets ($p53^{fl/fl}$;$Rb^{fl/fl}$;$p130^{fl/fl}$ (TKO) GEMM tumors, $Rb1^{fl/fl}$;$Trp53^{fl/fl}$;Lox-Stop-Lox[LSL]-$Myc^{T58A}$ (RPM) GEMM tumors, publicly available SCLC cell line data from CCLE.** CIBERSORT deconvolution provides proportions of SCLC subtypes in cell line data and tumor samples from their RNA sequencing signatures.
(CSV)

**S1 Fig. Comparing marginal likelihood-based posterior probability and information criteria probability estimation.** (A) Comparing posterior probability calculations from marginal

likelihood returned by nested sampling to posterior probabilities estimated by information criteria, for the 16-candidate model set and simulated data from the linear regression model selection example in Galipaud et al., 2014 [15]. Pearson correlation coefficient (r) is shown for each comparison. Comparison to posterior probability calculated from the BIC-estimated marginal likelihood, left. Comparison to AICc weights, right. (B) Comparing posterior probability calculations from marginal likelihood to posterior probabilities estimated by information criteria, for the SCLC analysis of 5,891 models compared to TKO data. Multimodel inference comparing candidate models to RPM and SCLC-A cell line data provide similar results. Pearson correlation coefficient (r) is shown for each comparison. Comparison to posterior probability calculated from the BIC-estimated marginal likelihood, left. Comparison to AICc weights, right.
(TIFF)

**S2 Fig. Prior probabilities values and schematics.** (A) Rate of a cell fate (division, death, or phenotypic transition) for $x$ ($v_{fate}$) can be calculated as a function of the population size of the effector cell $w$ (see S2 Text **Note A**) [1]. (B) Example calculation of division rate parameter prior for H841, representation of subtype Y, (see S1 Table) converting doubling times to "per day" units. (C) Division prior for subtype Y, (blue dashed line centered at the mean) as well as inter-subtype effect on division, whose mean is centered 5% lower (red dashed line; see S1 Table) with wider variance to account for more uncertainty in inter-subtype effects. (D) Example calculation and visualization of death rate parameter prior for Y (blue dashed line at mean) and inter-subtype effect on death (red dashed line at mean, 5% higher). (E) Example uniform transition prior, (see S1 Table) here showing N to Y transition; blue dashed line at baseline transition rate center, red dashed line at inter-subtype effect transition rate center. (F) Equilibrium assumption prior, representing $K_D K_x^{eq}$ in the equation (A). Each affected interaction has a unique $K_D K_x^{eq}$ prior, but all such priors have identical values (centered at 1000) before fitting. (G) Different model initiation hypotheses, where a model can be initiated by one or more subtypes (thick red outline) depending on the subtypes present in the topology. For an $n$-subtype topology, there are $2^n$-1 potential initial conditions. Here, the 4-subtype topology is shown, in a table representing all options for initial number of cells of each subtype (left) and in model schematics (right), with $2^4-1 = 15$ initial conditions. With equal prior probabilities, each hypothesis about which cell types initiate the tumor has a prior probability of 6.67%.
(TIFF)

**S3 Fig. Parameter prior distributions for all possible reactions in a candidate population dynamics model.** If a candidate model does not contain a reaction, for example a model with the topology A, N, and Y does not include A2 and thus will not include A2 division, death, or transitions to/from A2, then the rate parameter priors for A2-related reactions will not be included as a parameter prior for model fitting.
(TIFF)

**S4 Fig. Nested sampling's fitting results in better-fitting simulations than simulations using randomly selected parameter values.** Left, Data distribution, prior predictive distribution, and posterior predictive distribution for each dataset and all candidate models. Data is represented by a Beta distribution, bounded by zero and one, and used in the likelihood function input for Multinest (see Methods). Prior predictive distribution represents model simulations using parameters randomly drawn from the prior. Posterior predictive is generated by model simulations using best-fitting parameters returned by Multinest. Right, prior predictive distribution, simulation steady-state proportions using independently-sampled posterior

marginal parameters ("subtype independent post. predictive"), and simulation steady-state proportions using parameters sampled from the joint posterior distribution ("subtype joint posterior predictive", same as posterior predictive distribution on the left). See S3 Text for more detail and discussion related to these results. Data and predictive distributions for each dataset shown. (A) TKO, (B) RPM, (C) SCLC-A cell lines.
(TIFF)

**S5 Fig. Prior and posterior probabilities for tumor-initiating subtype hypotheses.** Hypothesis assessment of tumor-iniating subtypes, per dataset. Probability indicates the result of Bayes theorem using equivalent prior probabilities per initiating subtype, black dotted line (located at 6.67% probability that one of the initiation schemes in the x-axis best represents the data) and marginal likelihoods summed per initiation scheme. All topologies (Fig 5A) used in this analysis.
(TIFF)

**S6 Fig. Transition parameter rates vary in similar ways across datasets.** (A) Hypothesis assessment of model topologies per dataset, posterior probabilities based on all candidate models, with no filtering based on initiating subtype (see S4 Fig). Model topologies represented by images and corresponding numbers along the y-axis. (B) Comparison of phenotypic transition parameter posterior marginal distributions, BMA-weighted, per dataset, separated by topology. In 3- and 4-subtype topologies, distributions are further separated by hierarchical or non-hierarchical transition status. Bars indicate significance between samples from BMA parameter distributions at family-wise error rate (FWER) of 0.01, using one-way ANOVA plus Tukey HSD. Red bar: comparing N-to-Y rate with A2-to-Y rate, noted in the main text. Red arrowheads: higher A-to-A2 transition rate in 3-subtype TKO topology (A, A2, Y) compared to A-to-Y and A2-to-Y (noted in Fig 5B as well). Teal bar: comparing A2-to-N rate with N-to-A2 rate, noted in the main text. Teal arrowheads: higher A-to-N transition rate in 4-subtype RPM topology compared to A-to-Y and N-to-Y (noted in Fig 5B).
(TIFF)

## Acknowledgments

The authors would like to thank Sarah Groves, Michael Irvin, and Christine Lovly for insightful conversations and critical feedback on this work.

## Author Contributions

**Conceptualization:** Samantha P. Beik, Leonard A. Harris, Carlos F. Lopez.

**Data curation:** Samantha P. Beik.

**Formal analysis:** Samantha P. Beik.

**Funding acquisition:** Vito Quaranta, Carlos F. Lopez.

**Investigation:** Samantha P. Beik.

**Methodology:** Samantha P. Beik, Leonard A. Harris, Michael A. Kochen.

**Resources:** Julien Sage, Vito Quaranta, Carlos F. Lopez.

**Software:** Samantha P. Beik, Leonard A. Harris, Michael A. Kochen.

**Supervision:** Leonard A. Harris, Vito Quaranta, Carlos F. Lopez.

**Visualization:** Samantha P. Beik.

**Writing – original draft:** Samantha P. Beik, Carlos F. Lopez.

**Writing – review & editing:** Samantha P. Beik, Leonard A. Harris, Michael A. Kochen, Julien Sage, Vito Quaranta, Carlos F. Lopez.

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
