## [Editor Report · Decision Letter 0]

6 Jan 2023

Dear Dr. Lopez,

Thank you very much for submitting your manuscript "Unified Tumor Growth Mechanisms from Multimodel Inference and Dataset Integration" for consideration at PLOS Computational Biology.

As with all papers reviewed by the journal, your manuscript was reviewed by members of the editorial board. Before sending out the manuscript for peer review, I would like to request upload of higher resolution figures, or a single pdf manuscript including good resolution figures. The current pdf generated by the system is poor quality (apologies if this is system limitation), and I am afraid that the reviewers will be unhappy with this. 

We cannot make any decision about publication until we have seen the revised manuscript. Your revised manuscript is likely to be sent to reviewers for evaluation.

Sincerely,

Kiran Raosaheb Patil, Ph.D.

Section Editor

PLOS Computational Biology
---

## [Decision Letter · Decision Letter 1]

6 Mar 2023

Dear Dr. Lopez,

Thank you very much for submitting your manuscript "Unified Tumor Growth Mechanisms from Multimodel Inference and Dataset Integration" for consideration at PLOS Computational Biology.

As with all papers reviewed by the journal, your manuscript was reviewed by members of the editorial board and by several independent reviewers. In light of the reviews (below this email), we would like to invite the resubmission of a significantly-revised version that takes into account the reviewers' comments.

We cannot make any decision about publication until we have seen the revised manuscript and your response to the reviewers' comments. Your revised manuscript is also likely to be sent to reviewers for further evaluation.

Sincerely,

Pedro Mendes, PhD

Academic Editor

PLOS Computational Biology

Kiran Patil

Section Editor

PLOS Computational Biology

Reviewer's Responses to Questions

**Comments to the Authors:**

Reviewer #1: In their manuscript "Unified Tumor Growth Mechanisms from Multimodel Inference and Dataset Integration" Beik et al. introduce a mathematical/computational model selection framework (Bayes-MMI) and apply it to investigate different competing hypotheses about the inner workings of small cell lung cancer comprised of multiple interaction tumor cell populations.

The paper is well written and the methodology of Bayesian Model selection is an adequate approach to compare (and aggregate over) the zoo of thousands of possible mathematical models to describe SCLC dynamics. Moreover appropriate, scalable model selection is an important tool to deal with the ever increasing space of data and hypothesis in current biomedical research.

The structure of the manuscript could be improved for the sake of clarity. The main contribution is this new model selection method (Bayes-MMI) and it would deserve its own section in the "Results" (maybe together with some general model selection background). Currently the description of Bayes-MMI (and model selection in general) is interspersed and scattered over several Results-sections discussing mostly the SCLC biology and results.

*More importantly*, currently the manuscript is missing a few crucial mathematical details, in particular what data was fitted exactly, how the data was fitted, and if the data contains any information to actually distinguish those models.

Any biological conclusions may be highly dependent on those mathematical details.

I hence recommend a major revision to address those shortcomings in the current version of the manuscript. See below for details.

## Major

### Fitting to SCLC proportions

The authors use proportions of cell subtypes (Fig2) as data and their models are fit to that data.

1. From the Methods (l.526) it appears that the authors fit their models' steady state ($X(t=\\infty | \\theta)$) to this data. How this is done is a bit obscure: The authors calculate "alpha and beta using mean and variance of each (computational?!) dataset" (l.551). The authors evoke a Beta distribution $P(x|\\alpha_{sim}, \\beta_{sim})$ around their model's (normalized) steady state to calculate the likelihood of an observed datapoint $x$. $\\alpha_{sim}, \\beta_{sim}$ are related to the model's normalized steady state, i.e. $\\frac{\\alpha_i}{\\alpha_i+\\beta_i}$ = $X_i(\\infty| \\theta)$. **However it is unclear how the variance of the beta distribution is chosen or why a beta distribution is a reasonable choice.** As ODEs are deterministic, for a given set of parameters $\\theta$, the variance in steady state values is zero! The choice of a Beta distribution seems arbitrary. The authors are trying to match to proportions (data and simulation), I don't see an obvious connection to a Beta distribution. If the observed data were total counts instead of proportions, one could argue for a Dirichlet/Multinomial connection between model and data, with the variance in the data being explained by sampling. This is not the case here.

2. **Most importantly**: As the authors only use steady state data, I don't see how this information is enough to fit those models reliably, let alone do model selection. Similar papers (e.g. [10.1016/j.celrep.2018.11.088](https://doi.org/10.1016/j.celrep.2018.11.088)) use timecourse data with known initial conditions (via cell labeling) and yet still it's hard to distinguish models. Even simple population models (no interaction) will have identifiability issues: Due to the steady state nature of the data, it's impossible do disentangle growth and death rates. The compositional/proportional nature of the data adds to identifiability issues, hence making model selection tricky. Very different models (with all different kinds of interactions) could give the exact same steady state. Those interactions (e.g. one cell type inhibiting another cell type) would be detectable in correlations (whenever celltype A goes up, celltype B goes down) away from steady state (transient dynamics), but not in steady state. As the framework explicitly assumes independence (the likelihood is a product of beta-distributions), these correlations would be impossible to leverage. Given these concerns and some of the modelling choices (Beta distributions, see 1), it's unclear if the selected models actually explain the data better, or if models cant really be distinguished due to lack of data and model selection just picks the ones that satisfy some of those modeling artifacts.

## Minor

### Structure and mathematical details

A general introduction to Bayesian model selection and the Bayes-MMI method deserves it's own Results section independent of it's application to the CCLC data. Maybe a toy example. Most importantly, show the model-class that you're fitting: ODEs. Some mathematical notation might be helpful, also later in the manuscript: $\\frac{d X_i}{dt} = f_i(X1,\\ldots X_n | \\theta)$, $X_i$ being the cell types etc... explain the details of $f$, growth, death, interaction. I guess it's something like $f_i(X) = \\alpha_i + \\delta_i X_i + I(X_i, X_j)$ . How's that interaction term $I$ modeled? Hill kinetics? What are the initial conditions of the ODE (are they fitted)? More importantly how's the likelihood function $L(D|\\theta)$ defined, and what's the data $D$ in the first place. The authors should be much more explicit about the underlying math!

Clearly separating between new methodology and its application would make the manuscript more concise and avoid the current jumping back and forth between math and biology,

### Other

- What happens using this approach when faced with a multistable system (not in the data, but on the model side), e.g. two celltypes with self-activation and cross inhibition. Since the authors explore the universe of all (or at least a lot) of model topologies and parameters, one eventually must come across a multistable configuration

- Fig4A: node labels are much too small

- The model likelihood is a Beta-distribution around the models mean. How are the individual observations (bars in Fig2) incorporated into the likelihood? Product of likelihoods of each single observation. Or are the observations averaged and only the average proportion (per tumor type) enters the likelihood?

Reviewer #2: I enjoyed reading the main body of the text, and strongly support the approach taken to consider multiple models and use Baysian inference & model comparison – this is exactly how comparing models with data should be done, and the ranking of thousands of models is impressive. However, the novelty of this approach is overclaimed. Reading the introduction, one gets the impression that the authors were the first to apply such an approach, which is clearly not the case.

My main suggestions for revision are:

1. Rephrase the introduction with respect to novelty about the application of multimodel inference. The field of Bayesian model is well established and you are not the first to apply to this to biological models (and you refer to prior work (ref 44) that already takes the marginal likelihood approach).

2. Relating to the above, you show your BMMI to outperform AIC. A commonly used alternative to AIC is BIC, which can be derived as an approximation from the marginal likelihood (as can be readily read on Wikipedia). Would the BIC offer a computationally more efficient way to get the same results (as you don’t have to integrate the posterior distribution)?

3. How is likelihood computed (given that the model is deterministic)? The SI note refers to “least-squares likelihood function” (implying Gaussian errors?), whereas the methods mention a beta function (motivated by sampling).

4. Why not try and run inference on all 3 datasets together? How would this compare to the consolidated model in Fig 5e? It’s not obvious to me that they would be the same, and the former seems the more principled approach.

5. Can you show joint posterior distributions for (some of) the top models, to show whether some parameters are correlated/only identifiable in combination?

6. Could you make some concrete predictions of experiments that would be most informative to further distinguish between top ranked models, or to be most informative of your consensus model? E.g. by simulating from the model and determining where the predictions deviate most.

Minor points:

A. In line 78 you state you are interested in the case where hypotheses cannot be exhaustively enumerated, then in line 86 you state that you enumerate hypotheses…

B. Good visual summary of existing hypotheses in Fig 1, however Fig1 B-E is somewhat confusing to follow.

C. Fig S2 caption refers to Box 1, and in the bioRxiv preprint I can see boxes, but not in the version provided to me by the journal. (If the boxes are to be used: In Box 1C, is it a realistic assumption that signalling factor f is free after binding to x (rather than remaining bound to x*)?

D. Is it valid to use BMA to average posterior parameter distributions? (can you provide a reference?) Fig 4B is also hard to read, as Fig 5f

Reviewer #3: The manuscript addresses a stubborn challenge in computational systems biology and does so with an inventive and rigorous methodology that could have substantial impact to the field. They demonstrate it’s applicability using a challenging and medically important question with SCLC. Interpretation of highly complex analysis is clear throughout and the biological implications clearly highlighted.

The broad applicability of the method is challenged by the computational complexity of the approach, and it is demonstrated on a single biological question, but this is expected given the size of the challenge the authors are trying to overcome. Given increasing access to increasing computing power the methodology this is not likely to be a barrier to future utility in diverse biological areas.

Addressing the concerns below will ensure the broad applicability of the approach is clear to readers.

Major:

1. Can the authors clarify the paragraph starting on line 176, and whether each candidate model was chosen due to the potential to capture all the data in table 1. My concern is that some data in table 1 is based on a single reference, and one on unpublished correspondence. As such the reliability of each observation could be challenged by future studies. I would suggest including candidate models that explain all but 1 entry in table 1, where the non-included entry is either a single study or unpublished.

Obviously the time for running the workflow may preclude this analysis but the authors should address how incorrect data in table 1 would affect subsequent results. Any analysis that sheds some light on how each property described in Table 1 impacts the number of candidate models and subsequent results would be powerful in understanding how the literature impacts the outcome of the approach.

2. Can the authors explain the shape of the probability distribution in Figure S3. The “digital “ distribution such as diff_A_to_N_Baseline? These distributions suggest some parameters values are impossible at a specific value and possible for another one arbitrarily close.

a. Similarly the shape of the density distributions in Fig S4 should be explained. Subtype Y shows a bi/tri-modal distribution with areas of low density, based on the RPM data. Please describe how these arise. Does the SCLC-A cell line data and TKO data suggest the modes between [0.2,0.4] and [0.6,0.8] are not real as they are not recapitulated by different data sources?

3. While I know the analysis is not aiming to find a single best model, and small models have an “unfair” advantage, figure 4 suggest compelling evidence that topology 7 has some underlying truth to it, as the only topology that has very high probability in all datasets. Despite this the authors focus on topologies 1,2, and 4, arguing that 2-node topologies such as topology 7 do not capture all the subtypes required. Can the authors expand on why topology 7 was not taken forward? Why does their modelling not shed doubt on the experimental evidence that there is complex multi-subtype tumor composition, as a simple model fits the data best? Can the authors discuss why their approach might identify an unsuitable topology as a “winner” and how this might be avoided when others apply their technique? How could one identify topologies that might not align with experimental evidence if they have high likelihood in multiple datasets? How could the approach be adjusted to ensure an overly small model doesn’t “win”?

4. I do not understand how a “consolidated three-subtype topologies”, which produces a “unifying 4 subtype topology” (Figure 5E), can be a reliable unifying mechanism when the 4 subtype topology did poorly in Figure 4A (topology 1)?

Minor:

5. While “recalcitrance” is a term used in the SCLC field, but given the broad applicability of the methods described perhaps a term more broadly understood outside the cancer field would be appropriate? “treatment resistant” or just “poor survival”? At least for the abstract.

6. I would disagree that Akaike Information Criterion, (AIC) has had limited success as it has been used in countless papers to select the most appropriate model. Can the authors point to any examples where AIC failed to select the right model?

7. The introduction suggests that Bayes-MMI is most applicable when “models cannot be exhaustively enumerated”, but then to demonstrates the approach by exhaustively enumerating: “mechanistic hypotheses” to generate “thousands of candidate population dynamics models”. This suggests their test case may not be the best example. I would suggest rewording this text as it sets the wrong expectation.

8. NE is used (in the intro) before it is defined (in the results)

9. TFs in Figure 1 are too small to see.

10. Figure 1 B suggests that C(4,1)=1, surely this should be 4?

11. It is not clear how the media is conditioned in the bullet point in table 1, or what the exosomes are isolated from? Is it from a mixed SCLC culture or a particular subtype?

12. I do not agree with this sentence: “Therefore, our results indicate that the data used for model fitting has informed our knowledge about the system, because before nested sampling, all models are equally likely” The authors chose to make all models equally likely, which is correct methodologically, but they likely had intuition based on evidence that some were more likely than others. Almost any approach could move from equal likelihood to something non-equal so I would suggest either removing or rewording this claim.

13. The authors state that “we found that the fitted parameter

distribution outcome was dependent on choice of initiating subtype Fig S5” I actually find it remarkable how independent of initiating subtype almost all distributions are. I believe the need to narrow down based on initiating subtype can be justified without this extra step. Perhaps it should have been included in number 1.

14. Please make the letter’s in the model schematics clearer in figure 4A as the two node topologies all look identical and are only discriminated by tiny black on grey letters that are not legible when printed.

15. Rather than Fig 4B left, third-darkest blue can the authors indicate the important point with a red arrow.

16. Some of the references appear as internet sources when they are in fact published in journals.

- Simon Mitchell

**Have the authors made all data and (if applicable) computational code underlying the findings in their manuscript fully available?**

Reviewer #1: Yes

Reviewer #2: Yes

Reviewer #3: Yes

PLOS authors have the option to publish the peer review history of their article (what does this mean?). If published, this will include your full peer review and any attached files.

Reviewer #1: No

Reviewer #2: No

Reviewer #3: **Yes: **Simon Mitchell
---

## [Decision Letter · Decision Letter 2]

4 May 2023

Dear Dr. Lopez,

Thank you very much for submitting your manuscript "Unified Tumor Growth Mechanisms from Multimodel Inference and Dataset Integration" for consideration at PLOS Computational Biology. As with all papers reviewed by the journal, your manuscript was reviewed by members of the editorial board and by several independent reviewers. The reviewers appreciated the attention to an important topic. Based on the reviews, we are likely to accept this manuscript for publication, providing that you modify the manuscript according to the review recommendations.

Sincerely,

Pedro Mendes, PhD

Academic Editor

PLOS Computational Biology

Kiran Patil

Section Editor

PLOS Computational Biology

Reviewer's Responses to Questions

**Comments to the Authors:**

Reviewer #1: The revised version of their manuscript "Unified Tumor Growth Mechanisms from Multimodel Inference and Dataset Integration" addresses all the points I raised with the original manuscript. The modeling itself is introduced more explicitly and kept separate from the biological system under investigation. The structure of the manuscript is now much clearer. Thank you, I appreciate the effort!

The authors agree with my statement about the limited information content in their steady state data and mention that in the discussion, stating that cell-cell interactions are "uninformed by the data".

I think this might be an issue beyond cell-cell interactions (i.e. the abundance of a cell type influencing division/death/transition of another). Even cell-cell transitions could be unidentifiable. For example, take the model depicted in Fig.4C as ground truth (two cell types x and w, both dividing and dying, plus a unidirectional transition from x to w). Given just deterministic steady state data of this model, I don't see how one would be able to distinguish a model with a non-zero transition rate k_{x->w} from a model where both cell types are completely separated from each other (k_{x->w}== 0) unless the division and death rates are known/fixed. One would not be able to tell which topology is the correct one given the steady state data.

I would like the authors to briefly comment if this identifiabilty issue indeed extends to the transitions rates. Since the transition rates between the different cancer-cell subtypes are a major focus in the later part of the manuscript, this should to be clarified.

Minor adjustments:

- Fig4B: minus instead of plus signs, as death and transition decrease the amount of x

Reviewer #3: The authors have substantial revised the manuscript in response to the comments by all reviewers and it is much improved.

Particularly the promotion of a "toy" example to the main text provides a compelling demonstration of the technique.

All my comments have been adequetly addressed and I think the study is now more likely to be applied more broadly by the systems biology community to a variety of biological disciplines.

**Have the authors made all data and (if applicable) computational code underlying the findings in their manuscript fully available?**

Reviewer #1: Yes

Reviewer #3: Yes

PLOS authors have the option to publish the peer review history of their article (what does this mean?). If published, this will include your full peer review and any attached files.

Reviewer #1: No

Reviewer #3: **Yes: **Simon Mitchell

Figure Files:

Data Requirements:

Reproducibility:

References:

---

## [Editor Report · Decision Letter 3]

25 May 2023

Dear Dr. Lopez,

We are pleased to inform you that your manuscript 'Unified Tumor Growth Mechanisms from Multimodel Inference and Dataset Integration' has been provisionally accepted for publication in PLOS Computational Biology.

Best regards,

Kiran Raosaheb Patil, Ph.D.

Section Editor

PLOS Computational Biology

---

## [Editor Report · Acceptance letter]

28 Jun 2023

PCOMPBIOL-D-22-01877R3 

Unified Tumor Growth Mechanisms from Multimodel Inference and Dataset Integration

Dear Dr Lopez,

I am pleased to inform you that your manuscript has been formally accepted for publication in PLOS Computational Biology. Your manuscript is now with our production department and you will be notified of the publication date in due course.

With kind regards,

Bernadett Koltai
